



# Increasing Daily Extreme and Declining Annual Precipitation in Southern Europe: A Modeling Study on the Effects of Mediterranean Warming

Alfonso Senatore[1], Luca Furnari[1], Gholamreza Nikravesh[1], Jessica Castagna[1], and Giuseppe Mendicino[1]

[1]Department of Environmental Engineering, University of Calabria, Rende 87036, Cosenza, Italy

**Correspondence:** Alfonso Senatore (alfonso.senatore@unical.it)

**Abstract.** Understanding the evolving patterns of intense rainfall in the Mediterranean under climate change is an urgent challenge. An analysis of the annual total and maximum one-day precipitation from 1955 to 2023 performed with the ERA5-Land dataset over the EURO-CORDEX domain reveals emerging patterns of contrasting trends along much of the northern Mediterranean coast, with heavy precipitation events increasing and total annual rainfall decreasing. An independent investigation on a ground-based dense monitoring network in southern Italy confirms the results. We focus on this representative sub-region of the study area to examine in detail the role of sea-atmosphere-orography interactions, particularly the impact of increasing sea surface temperature (SST), in enhancing heavy precipitation despite overall drying. Twenty consecutive precipitation events identified in a particularly intense rainy season (September-December 2019) are reproduced at a convection-permitting resolution (2 km) using the Weather Research and Forecasting (WRF) model with ERA5 reanalysis boundary conditions. Then, two scenarios are tested: one with past SST levels approximating 1980 and another with future SST increases in line with end-of-century Shared Socioeconomic Pathways (SSPs) projections. WRF simulations thoroughly describe cyclone tracks and precipitation patterns, showing that increased SST boosts the frequency of heavy rainfall events overland, though peak intensities remain mostly unchanged because the highest precipitations occur over the sea. The study demonstrates the unique capability of high-resolution, convection-permitting analyses to capture complex processes in orographically challenging regions and contributes to clarifying the seemingly contradictory trend of rising daily precipitation extremes despite falling annual precipitation totals in Southern Europe.

## 1 Introduction

Last century's observations confirm that the global warming trend has not had a uniform impact worldwide on the local climates and hydrological cycles. The carbon dioxide-induced warming is more emphasized in the Arctic pole, especially in winter (Stone, 2021), but severe implications on weather can also be detected in the mid-latitudes of the Northern Hemisphere, where cold anomalies due to outbreaks of Arctic cold air are detected (Kömüşcü and Oğuz, 2021) and the expansion of the Hadley Cell is slightly faster than in the Southern Hemisphere (Xian et al., 2021). Generally, the atmospheric circulation of the thermally closed Hadley Cell pushes the warmer fluid from the equator polewards, while cold fluid is pushed towards the subtropics and equatorward. During the last decades, as confirmed by chemical metrics, the Hadley Cell shows a poleward





trend, which may cause more frequent drought at higher latitudes (Xian et al., 2021). For instance, the Mediterranean area, as well as the Middle East, Western Pacific, and Asian monsoon regions, are hit by more frequent heatwave events due to the expansion of the Hadley Cell, causing a further Sea Surface Temperature (SST) increase, already driven by global warming (Pastor et al., 2020).

     The annual warming rate recorded in Mediterranean SST is about 0.035 °C/year, corresponding to a rate increase of about

1.3 °C from 1982 to 2019 (Pastor et al., 2020; Mohamed et al., 2019) for the whole basin, with a maximum rate in the Eastern basin (0.040 °C/year), and lower rates related to the western and central basin (0.035 °C/year and 0.031 °C/year, respectively) (Pastor et al., 2020; Sannino et al., 2022). In the last two years, the recorded Mediterranean Sea SST was the highest since the 1950s (Cheng et al., 2024, 2025).

     Moreover, the Mediterranean region is generally hit by significant cyclonic activity, which during the last 4 decades (1979-

2018) increased by 40% more (Aragão and Porcù, 2022), favoring the formation of tropical-like cyclones/hurricanes, called Medicanes (Miglietta and Rotunno, 2019), characterized by destructive winds and torrential rainfall (Zhang et al., 2021). For instance, frequent extreme precipitation events caused by Medicanes were recorded in Europe in the period 1979-2017, among which over 20 were in Italy, followed by France, Croatia, Serbia, and Greece (Zhang et al., 2021).

     From the point of view of physics, cyclogenesis processes are influenced by thermodynamical and dynamical factors. Ther-

modynamically, the SST warming, which defines the air-sea heat flux, contributes to lead larger evaporation and more moisture content, especially in the lower Planetary Boundary Layer (PBL) and in a minor part in the upper PBL (Khodayar et al., 2021; Sun and Wu, 2022). Additionally, the effects of increased moisture may trigger dynamically with the orographic lifting, causing deep convection, an increased Convective Available Potential Energy (CAPE), and as the final result, extreme precipitation events (Müller et al., 2024; Ricchi et al., 2023; Pfahl et al., 2017; Meredith et al., 2015).

The increasing trend of the Mediterranean SST is predicted to continue. Future scenario projections of SST suggest additional warming. As an example, under RCP8.5, an increase of over 3 °C is predicted (+3.31 °C for 2081-2100 related to the upper layer 0-100 m for the whole Mediterranean Basin, +2.98 °C for the western basin and +3.50 °C for the eastern basin) (Sannino et al., 2022). On the other hand, the spatial patterns of future precipitation trends are more heterogeneous and complex to detect. The great challenge of deciphering how global warming will affect rainfall at local scales (Chadwick, 2017)

is particularly tough in regions with transitional climate regimes, such as the Mediterranean. In the subtropics, a large-scale precipitation decline is generally projected (He and Soden, 2017). However, the indications about how this phenomenon will affect local areas can be different depending on whether one relies on analyses at low spatial resolutions such as GCMs (Pfahl et al., 2017), or gradually increasing through climatic downscaling (Tramblay and Somot, 2018; Hosseinzadehtalaei et al., 2020; Reale et al., 2022; Matte et al., 2022) up to convection-permitting resolutions (Müller et al., 2024).

Several studies evaluated the SST's strong influence on cyclonic activity in the Mediterranean as well as the Medicanes. Basically, as intuitively expected, with a negative variation of SST, the cyclones are weaker, while in warmer SST conditions, precipitations increase (Miglietta et al., 2011; Meredith et al., 2015; Ricchi et al., 2017, 2023; Varlas et al., 2023). However, even slight uniform variations of SST (± 0.5 K) can influence the precipitation patterns in some particular synoptic conditions (Senatore et al., 2014). Similarly, it has been shown that a larger variation of SST from -2 °C to +2 °C strongly affects extreme




events as the Ianos cyclone occurred in 2020 (Varlas et al., 2023), producing a variation of precipitation intensity from -56% to +44% and influencing the tracks and the landfall location. Also, the high-resolution representation of the SST pattern highlights a weak but not negligible impact on the operational forecasting results in many cases (Ricchi et al., 2017; Senatore et al., 2020b), contributing to the overall uncertainty along the modeling chain (Senatore et al., 2020a).

The climate of Southern Europe, whose coastline broadly faces the Mediterranean Sea, is influenced on the one hand by

the SST and on the other hand by an often locally complex orography with steep mountains opposing to atmospheric water transport, both conditions rendering this region prone to extreme precipitation events characterized by strong air-sea-orography interactions (Berthou et al., 2016; Senatore et al., 2020b). In this study, we first analyzed current trends of intense precipitation considering maximum one-day observations for the whole Euro-Mediterranean region. To this aim, we used the data provided by the ERA5-Land dataset (Muñoz-Sabater et al., 2021) over the pan-European EURO-CORDEX domain (27° to 72° N and

22° W to 45° E, Jacob et al. (2014)). These trends were compared with total annual precipitation trends to disentangle possible contrasting behaviors of the two variables. Then, the robustness of the results achieved with the large-scale reanalysis was tested with local observations from a dense ground-based monitoring network available in the southernmost part of the Italian peninsula, located almost at the geographical center of the Mediterranean Sea. The comparison showed general agreement between the two datasets and proved the regional significance of this sub-region. Then, to underscore the current and future

influence of SST on the dynamics and intensity of precipitation events in Southern Europe, we focused on this representative sub-region, considering twenty events occurred during a particularly intense rainy season (from September to December 2019) and reproducing them using the WRF (Weather Research and Forecasting, Skamarock et al. (2021)) model with boundary conditions provided by ERA5 reanalysis for two nested domains. Successively, while preserving the other boundary conditions, we varied the SST lower boundary conditions for two scenarios: a past scenario with a decrease in SST equivalent to the SST

recorded approximately in 1980 and a future scenario with an increase reasonably representative of the projections for the end of this century according to selected Shared Socioeconomic Pathways (SSPs) scenarios. The overall analysis provided a comprehensive insight into the evolution of heavy precipitation intensity and frequency in the northern coastal Mediterranean and its correlation to sea surface warming.

## 2 Data and Methods

### 2.1 Datasets and study area

This study used several datasets to investigate historical trends in precipitation and the influence of increasing SST on heavy precipitation events.

For the trend analysis at the euro-Mediterranean level, the ERA5-Land data were used for the 69-year period 1955-2023. The ERA5-Land dataset is an enhanced global land dataset at $0.1°$ resolution derived from the 5th generation of European

ReAnalysis (ERA5), produced by the European Centre for Medium-Range Weather Forecasts (ECMWF) in the framework of the Copernicus Climate Change Service (C3S) of the European Commission (Muñoz-Sabater et al., 2021). ERA5-Land data were chosen because of their relatively high resolution and the concurrent availability of coherent precipitation and temperature





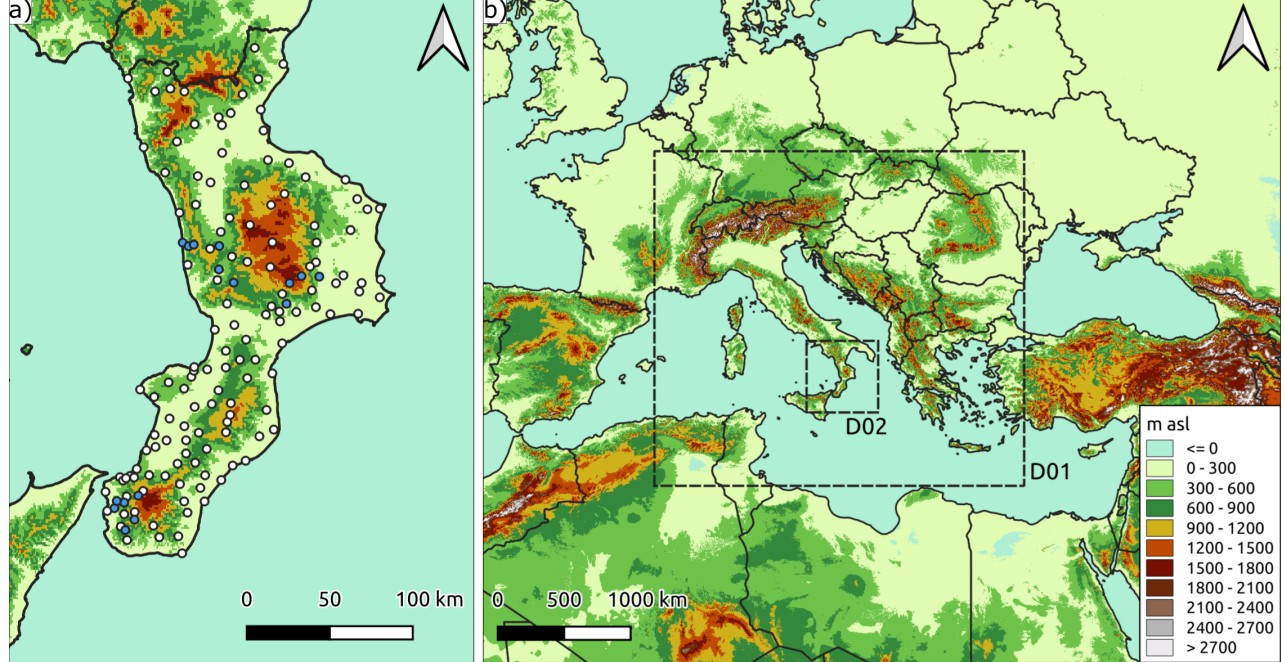

**Figure 1.** a) Overview of the Calabria Region with the administrative borders and the location of the 134 historical precipitation gauges used for the trend analysis (white dots). In addition, 16 other gauges were used for the 2019 spatial precipitation distribution (blue dots); b) Topographic map of the Central Mediterranean area showing the two outermost (D01) and innermost (D02) WRF domains.

data. Furthermore, ERA5-Land proved to be generally reliable in the selected domain (Muñoz-Sabater et al., 2021; Cammalleri et al., 2022; Vanella et al., 2022; Gomis-Cebolla et al., 2023; Ippolito et al., 2024).

The results based on the ERA5-Land dataset were compared at the local scale, considering the southernmost peninsula of Italy, which corresponds to the administrative region of Calabria. Calabria Region (Fig. 1a), covering an area of 15080 km$^2$, lies between $37°54'$ and $40°09'$N and $15°37'$ and $17°13'$E. The deep influence of the surrounding sea on the atmospheric conditions, combined with the complex local orography, makes the Calabrian climate very heterogeneous, with sharp transitions from humid to dry areas (Mendicino and Versace, 2007; Senatore et al., 2020a). The peculiar physical characteristics of
the territory not only make it prone to drought risk (generally higher in the eastern part) but contribute to the development of extreme and damaging hydrometeorological events on both eastern (Ionian) and western (Tyrrhenian) coasts, especially during the late summer until the late fall season when considerable amounts of precipitation in a short time cause severe floods with economic and social damages, including fatalities (e.g., Llasat et al. (2013); Petrucci et al. (2018); November 2015, Avolio and Federico (2018); November 2016, Senatore et al. (2020a); November 2019, Furnari et al. (2022)). Daily precipitation data were
collected in Calabria from 134 gauges (roughly a gauge per 110 km$^2$) of the regional monitoring network (Fig. 1a). Techniques like linear correlation process and spline were applied to estimate missing values or reconstruct detected outliers.



The September – December period of 2019 was selected as a case study to assess the sensitivity of the events to varying SST boundary conditions because it was rainier than usual in southern Italy and characterized by two relevant extreme precipitation events (from 11[th] to 13[th] of November, Marsico and Rotundo (2019) and from 23[rd] to 25[th] of November, Fusto et al. (2019))

that affected the whole region, especially the Ionian (eastern) side. For this period, it was possible to benefit from new gauges, increasing the size of the ground-based observations to 150 gauges (Fig. 1a). Precipitation data were spatially interpolated through the inverse distance weighting (IDW) at the same resolution of the atmospheric model to allow easier comparison with its spatially distributed output.

We used several datasets to analyze the observed and projected SST values of the Mediterranean Sea to assess a reason-

able range of variations of the atmospheric model's boundary conditions. As observed data, the 5[th] generation ECMWF reanalysis, i.e., the monthly SST ERA5 reanalysis from 1980 to 2023 combining global model data with observations sampled globally, was selected. We also used the Copernicus Marine Service (CMEMS) observations (SST-GLO-SST-L4-REP-OBSERVATIONS-010-011, ID: ESACCI-GLO-SST-L4-REP-OBS-SST, C3S-GLO-SST-L4-REP-OBS-SST) on a daily scale from 1982 to October 2022, based on the reprocessing analysis of satellite and in-situ data. The spatial extensions of the

analyzed region correspond to the Mediterranean region adopted by the IPCC Atlas (30° to 45° N and -10° to 40° E).

Regarding SST projections, we used the internationally coordinated Coupled Model Intercomparison Project Phase 6 (CMIP6), providing 22 GCM (Global Circulation Model) results (for a comprehensive overview of the dataset, the reader is referred to Table 1). The projections selected belong to two different Shared Socioeconomic Pathways (SSPs, Meinshausen et al. (2020)) scenarios:

– the medium-high emission scenario, i.e., SSP3-7.0 (called Regional Rivalry) characterized by high GHG emissions and $CO_2$ double around current levels until the end of the century;

– the worst scenario, SSP5-8.5 (named Fossil-Fueled Development - Taking the Highway) with very high GHG emissions and $CO_2$ that roughly triple from current levels by 2075.

Although the likelihood of these intermediate to very high warming scenarios is debated (e.g., Schwalm et al. (2020); Haus-

father and Peters (2020)), they were taken into account to highlight as much as possible the effects of climate change on SST and indirectly on precipitation events driven by sea surface-atmosphere interaction.

## 2.2 Trend analysis

Non-parametric trend tests like the Mann-Kendall test (identifying significance at a 5% level) and Sen's slope estimator test (determining the trend slope per year) were employed to analyze long-time hydro-climatic series of annual precipitation (PRCP-

TOT) and maximum one-day precipitation (RX1day). Mann (1945) formulated a non-parametric trend detection test, whereas the test statistic distribution was given by Kendall (1948) for turning point and non-linear trend assessment. This test aims to realize how independently and randomly the data are ordered by comparing the rank order of observations with their time order, considering the null hypothesis of the non-trend and serial structure of correlations (Hamed and Rao, 1998). Concerning the



**Table 1.** List of datasets (IDs 0_1 and 0_2) and GCMs (IDs 1-22) considered for assessing observed/estimated historical and estimated future SST evolution in the Mediterranean Area. GCMs data have been provided by Instituto de Física de Cantabria (IFCA)

| ID | Complete Name | Institution | Model | Variant | Reference |
|---|---|---|---|---|---|
| 0_1 | ERA5 Reanalysis | ECMWF | Reanalysis | | Hersbach et al. (2020) |
| 0_2 | SST_GLO_SST_L4_REP_OBSERVATIONS_010_024 | CMEMS | Observation | | (CMEMS) (2023) |
| 1 | ACCESS-CM2_CSIRO-ARCCSS_r1i1p1f1 | CSIRO-ARCCSS | ACCESS-CM2 | r1i1p1f1 | Dix et al. (2023) |
| 2 | ACCESS-ESM1-5_CSIRO_r1i1p1f1 | CSIRO | ACCESS-ESM1-5 | r1i1p1f1 | Ziehn et al. (2023) |
| 3 | AWI-CM-1-1-MR_AWI_r1i1p1f1 | AWI | AWI-CM-1-1-MR | r1i1p1f1 | Semmler et al. (2023) |
| 4 | BCC-CSM2-MR_BCC_r1i1p1f1 | BCC | BCC-CSM2-MR | r1i1p1f1 | Xin et al. (2023) |
| 5 | CAMS-CSM1-0_CAMS_r2i1p1f1 | CAMS | CAMS-CSM1-0 | r2i1p1f1 | Rong (2023) |
| 6 | CESM2-WACCM_NCAR_r1i1p1f1 | NCAR | CESM2-WACCM | r1i1p1f1 | Danabasoglu (2023) |
| 7 | CMCC-CM2-SR5_CMCC_r1i1p1f1 | CMCC | CMCC-CM2-SR5 | r1i1p1f1 | Lovato and Peano (2023) |
| 8 | CNRM-CM6-1-HR_CNRM-CERFACS_r1i1p1f2 | CNRM-CERFACS | CNRM-CM6-1-HR | r1i1p1f2 | Voldoire (2023a) |
| 9 | CNRM-CM6-1_CNRM-CERFACS_r1i1p1f2 | CNRM-CERFACS | CNRM-CM6-1 | r1i1p1f2 | Voldoire (2023b) |
| 10 | CNRM-ESM2-1_CNRM-CERFACS_r1i1p1f2 | CNRM-CERFACS | CNRM-ESM2-1 | r1i1p1f2 | Seferian (2023) |
| 11 | CanESM5_CCCma_r1i1p1f1 | CCCma | CanESM5 | r1i1p1f1 | Swart et al. (2023) |
| 12 | EC-Earth3_EC-Earth-Consortium_r1i1p1f1 | EC-Earth-Consortium | EC-Earth3 | r1i1p1f1 | (EC-Earth) |
| 13 | FGOALS-g3_CAS_r1i1p1f1 | CAS | FGOALS-g3 | r1i1p1f1 | Li (2023) |
| 14 | GFDL-ESM4_NOAA-GFDL_r1i1p1f1 | NOAA-GFDL | GFDL-ESM4 | r1i1p1f1 | Krasting et al. (2023) |
| 15 | IITM-ESM_CCCR-IITM_r1i1p1f1 | CCCR-IITM | IITM-ESM | r1i1p1f1 | Panickal et al. (2023) |
| 16 | INM-CM5-0_INM_r1i1p1f1 | INM | INM-CM5-0 | r1i1p1f1 | Volodin et al. (2023) |
| 17 | IPSL-CM6A-LR_IPSL_r1i1p1f1 | IPSL | IPSL-CM6A-LR | r1i1p1f1 | Boucher et al. (2023) |
| 18 | MPI-ESM1-2-HR_MPI-M_r1i1p1f1 | MPI-M | MPI-ESM1-2-HR | r1i1p1f1 | Jungclaus et al. (2023) |
| 19 | MPI-ESM1-2-LR_MPI-M_r1i1p1f1 | MPI-M | MPI-ESM1-2-LR | r1i1p1f1 | Wieners et al. (2023) |
| 20 | NorESM2-LM_NCC_r1i1p1f1 | NCC | NorESM2-LM | r1i1p1f1 | Seland et al. (2023) |
| 21 | NorESM2-MM_NCC_r1i1p1f1 | NCC | NorESM2-MM | r1i1p1f1 | Bentsen et al. (2023) |
| 22 | UKESM1-0-LL_MOHC_r1i1p1f2 | MOHC | UKESM1-0-LL | r1i1p1f2 | Tang et al. (2023) |

slope estimator test, instead of the least squares estimator of a regression coefficient, the unbiased point estimator was defined
based on the median of the set of slope joining pairs of points by Sen (1968).

Finally, a rectangular two-axes coordinate system can be considered to better portray the compound trend investigation of the two variables. In this system, four zones can be defined according to the concordant or discordant, positive or negative behavior of the two variables. Therefore, having PRCPTOT trend and RX1day trend values in the X and Y axes, respectively, in the first zone (i.e., quadrant I), both coordinates are positive (i.e., the trend of both indices is positive). In contrast, in quadrant III, both
are negative.

## 2.3 WRF downscaling and validation

Weather simulations over the Calabrian region were performed using WRF (Skamarock et al., 2021) (Weather Research & Forecasting) V4.1 limited area model. Two one-way nested domains were used (see Fig. 1b): the outermost (D01) centered on the Italian peninsula, with a horizontal resolution of about 10 km ($33.04 - 49.85°$N, $3.59 - 28.59°$ E, thus producing 187×205



grid points); the innermost (D02) domain centered on the Calabrian region, with 2 km as horizontal resolution $(37.10 - 40.87°$ N, $13.88 - 18.71°$ E, 200×200 grid points). Both domains extended on 44 vertical atmospheric layers, up to 50 hPa, and 4 soil layers. The time step size was set to 60 s and 12 s for the D01 and D02 domains, respectively. The physical schemes configuration (Table 2) is the same adopted by Avolio et al. (2019), with the replacement of NOAH-MP instead of NOAH as the Land Surface Model. This configuration runs operationally for weather forecasting as an online service managed by the

University of Calabria since 2020 (https://cesmma.unical.it/cwfv2/).

**Table 2.** WRF physical schemes adopted.

| Component | Scheme | References |
|---|---|---|
| Microphysics | New Thompson | Thompson et al. (2008) |
| PBL | MJY | Janjić (1994) |
| Longwave | RTTM | Mlawer et al. (1997) |
| Shortwave | Goddard | Matsui et al. (2020) |
| Land Surface Model | NOAH-MP | Niu et al. (2011) |
| Cumulus | Tiedke (only D01) | Tiedtke (1989) |
| SST | sst_skin | Zeng and Beljaars (2005) |

The initial and boundary conditions were provided by ERA5 using the pressure levels mode; specifically, the following levels were used: 1000, 950, 900, 850, 800, 700, 600, 500, 400, 300, 200, 150, 100, and 50 hPa. SST boundary conditions were modified (increased or decreased) in different experiments according to the analysis of the GCM projections. The update frequency of the boundary conditions was set to 3 hours. The simulations started on the 1[st] of September 2019 and went on

until the 1[st] of January 2020.

The model performances were evaluated by adopting two classical weather forecast indices: the Fractional Skill Score (FSS) and the Critical Success Index (CSI), also denoted as Threat Score (TS) (Wilks, 2006). The FSS (Equation 1) ranges from 0 (mismatch) to 1 (perfect match) and highlights how the spatial pattern forecasted could match with respect to the observed one. Assuming a square-shaped neighborhood of length (size) $n$, $FSS_n$ is given by:

$$FSS_n = 1 - \frac{\frac{1}{N}\sum_{i=1}^{N}\left(P_f - P_o\right)^2}{\frac{1}{N}\left[\sum_{i=1}^{N}P_f^2 + \sum_{i=1}^{N}P_o^2\right]}$$
(1)

In the above equation, $N$ is the number of windows in the domain, and $P_f$ and $P_o$ are boolean values indicating if the pixel value is higher or lower than the threshold related to the observation. We applied the FSS by considering the events identified according to the procedure described in the next section 2.4 and using the 95[th] percentile on the cumulative precipitation observed as the threshold. Moreover, according to Senatore et al. (2020a), we calculated the FSS only for the Calabrian

peninsula, in which we collected and interpolated spatially highly reliable ground-based data, masking the sea.





The CSI (Equation 2) quantifies the fraction of observed and forecasted events that are correctly predicted by varying the rainfall thresholds. The index varies between 0 (no skill) and 1 (perfect skill):

$$CSI = \frac{hits}{hits + misses + false\ alarms} \tag{2}$$

We considered the observed and predicted precipitation cell by cell during the events to build up the contingency table by varying the thresholds, whose selected values were 0.2, 0.5, 1, 2, 5, 10, 20, and 30 mm.

## 2.4  Space and time events identification in the period September-December 2019

During the selected period for WRF simulations, the spatial average of the simulated daily precipitation over the inner domain with the actual SST boundary conditions was used to identify and label the events that occurred. Each event was assumed to start the day before the precipitation began and continue until the day after its end, so each event was considered as standing alone with respect to others.

Finally, the precipitation patterns for the current SST and modified SST scenarios were evaluated. In particular, for each event and given SST boundary condition, the barycenters of the precipitation patterns were calculated considering all the pixels in which the amount of the accumulated precipitation during the event exceeded the 95$^{th}$ percentile:

$$x = \frac{\sum_{i=1}^{N} Prec(i,j) * i}{\sum_{i=1}^{N} Prec(i,j)} \tag{3}$$

$$y = \frac{\sum_{j=1}^{N} Prec(i,j) * j}{\sum_{j=1}^{N} Prec(i,j)} \tag{4}$$

In the above equations, $i$ and $j$ indicate the indices of each of the $N$ pixels exceeding the 95$^{th}$ percentile threshold, while $Prec(i,j)$ is the accumulated precipitation value in the pixel with coordinates $(i,j)$. Finally, $x$ and $y$ are the coordinates of the barycenter of the precipitation event.

## 3  Results and Discussion

### 3.1  Trend Analysis

The Sen's slope and Mann-Kendall trend test results for PRCPTOT and RX1day over the EURO-CORDEX domain are shown in Figure 2(a and b). In general, eastern and southern regions of Europe are experiencing decreasing trends of PRCPTOT, while the trend for central and northern territories is positive (Fig. 2a). Furthermore, the PRCPTOT trend is more significant in the North. On the other hand, the RX1day trend is more scattered but overall slightly increasing, with approximately an average of 0.1 mm/year, resulting in more probable flooding challenges across Europe (Fig. 2b). The areas with significant trends are also more widespread in space, though more present in the North. These results are largely consistent with previous literature. The




6th IPCC Assessment Report (Masson-Delmotte et al., 2021) highlights significantly increased observed heavy precipitation in northern and central Europe and a lower agreement in the type of change for the Mediterranean area. In contrast, a significant increase was observed in agricultural and ecological drought in the Mediterranean and Western and Central Europe, with low
agreement for the trend sign in other European areas.

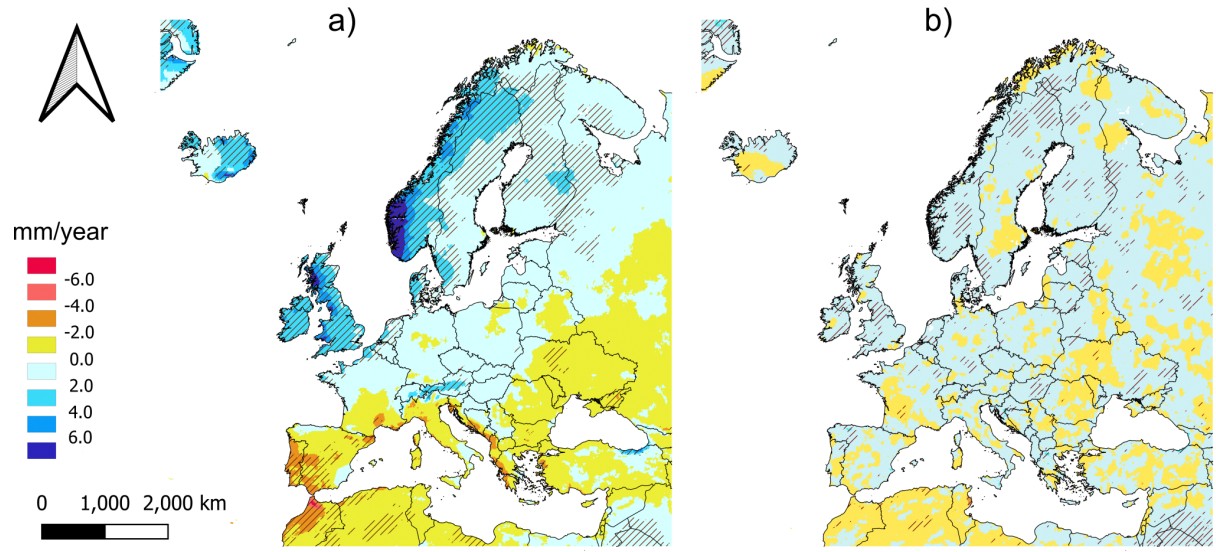

**Figure 2.** Mann-Kendall and Sen's slope (mm/year) tests for a) PRCPTOT and b) RX1day over the EURO-CORDEX domain, from 1955 to 2023. Dashed areas highlight significant trends.

The assessment of combined trends is particularly interesting. Figure 3 presents the spatial distribution of the areas falling in the four quadrants of a two-dimensional domain where the horizontal axis represents the PRCPTOT trend, and the vertical axis represents the RX1day trend. 68706 pixels out of about 143,000 (48%) are classified in the first quadrant (Zone I), where both PRCPTOT and RX1day trends are positive. Similarly, 30630 (21%) and 31078 (22%) pixels are placed in the second and third
quadrants (Zones II and III), respectively, while the number of pixels belonging to the fourth quadrant (Zone IV) is the least (9%). Generally, the central and northern parts of Europe are characterized by an increasing trend of both extreme and mean precipitation (Zone I). In contrast, the southern (Mediterranean) regions of Europe and, to some extent, the eastern regions are encountering decreasing PRCPTOT trend (hence, higher drought risk) and increasing RX1day trend, hence, higher flood risk (Zone II). Other eastern European areas, as well as southern Mediterranean coasts, are experiencing generalized decreasing
trends (Zone III).

The results achieved with the ERA5-Land reanalysis dataset were compared locally with the ground-based observations in the Calabrian peninsula. Of the 134 stations considered, 29 are recognized with PRCPTOT significant negative trends (Fig. 4a), while 4 stations have a significantly positive trend. In addition, 8 stations with significantly positive trends are found concerning RX1day (Fig. 4b, only 3 stations with significant negative trends, in this case). These results reveal consistent outcomes with





**Figure 3.** Four zones of compound trends of PRCPTOT and RX1day at annual scale over the EURO-CORDEX domain from 1955 to 2023. Dashed areas indicate that both PRCPTOT and RX1day trends are significant.

those of previous studies. In particular, Prete et al. (2023) found increasing daily rainfall extremes for 50- and 100-year return periods, Avino et al. (2024) pointed out generally increasing trends for daily and sub-daily rainfall durations, and Caloiero et al. (2020, 2021) realized a decreasing trend for the annual rainfall. On the contrary, Caloiero et al. (2017) asserted a decreasing trend of the higher daily rainfall categories, but with a dataset stopping in 2006.

     The four zones of combined PRCPTOT and RX1day trends are depicted in Figure 5. Nearly half of the stations (55, corre-
sponding to 41.0%) are dotted in Zone (quadrant) II, facing increasing drought and extreme precipitation risk. Then, Zone III (both decreasing PRCPTOT and RX1day) is noticed with 50 stations (37.3%), whereas 23 stations (17.2%) are identified in Zone I. Finally, only 6 stations (4.5%) are placed in Zone IV. Only the stations with both significant PRCPTOT and RX1day





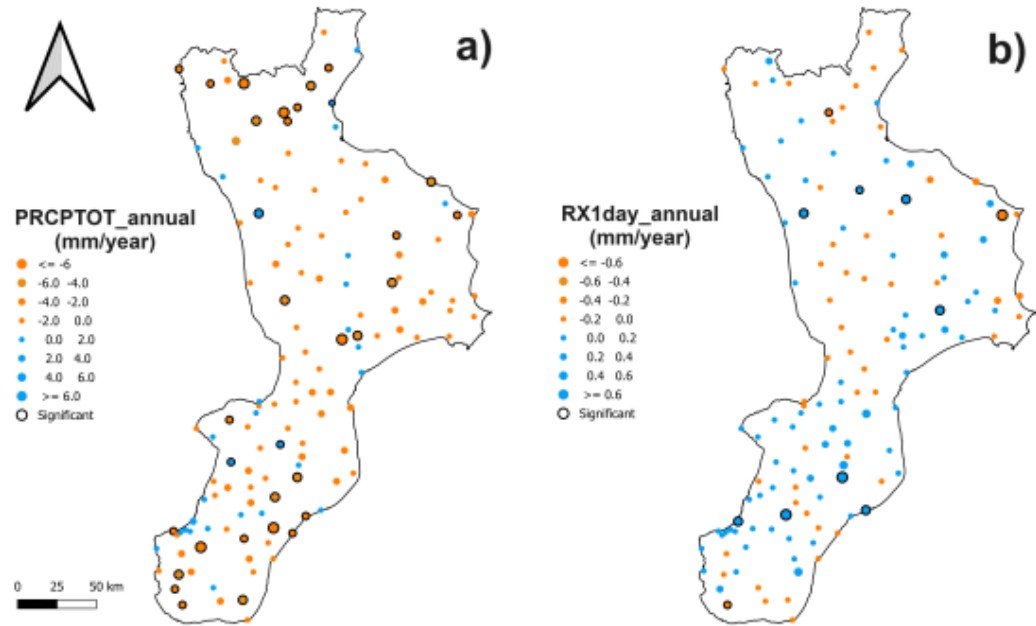

**Figure 4.** Mann-Kendall and Sen's slope tests for a) PRCPTOT, b) RX1day over Calabria from 1955 to 2023.

trends are highlighted as "significant" in the maps. They are very few, with a maximum of three stations in Zone III. Overall, results indicate that Calabria mainly confronts higher water stress risk, with PRCPTOT reduction detected in 105 stations
(78%), mostly combined with temperature increase (not shown). Nevertheless, the risk of extreme precipitation events is not decreasing in the majority of the region (58% of the stations analyzed), with broad clusters of increased compound risk in the south, east, and northwest of the area (Zone II in Figure 5).

The comparison between the large-scale reanalysis dataset and the local-scale ground observations demonstrates that the ERA5-Land dataset can catch the contrasting daily and annual precipitation trends in Calabria and further strengthens the
reliability of the results achieved along the entire northern Mediterranean coast. Based on a reanalysis, this dataset relies on observations indirectly, providing a spatially homogeneous level of accuracy. Other spatially distributed datasets directly based on ground data supply biased information in zones where the monitoring network is not dense enough. For example, the stations feeding the E-OBS dataset (Cornes et al., 2018) are rather sparse in southern and eastern Europe (Cammalleri et al., 2024). Also, concerning the hydrological impact, the findings of Stahl et al. (2012) and Blöschl et al. (2019) about decreasing
regional trends of river flood discharges in southern Europe are contradicted by Prete et al. (2023) and Avino et al. (2024) local-scale findings about daily precipitation trends probably because in the studies analyzing streamflow data very sparse observations are available for this region. Finally, a very recent study based on a comprehensive station-based dataset for the whole Mediterranean region (Vicente-Serrano et al., 2025) confirms a general decreasing trend for annual precipitation in the period 1951-2020, attributing it to internal variability of atmospheric dynamics.





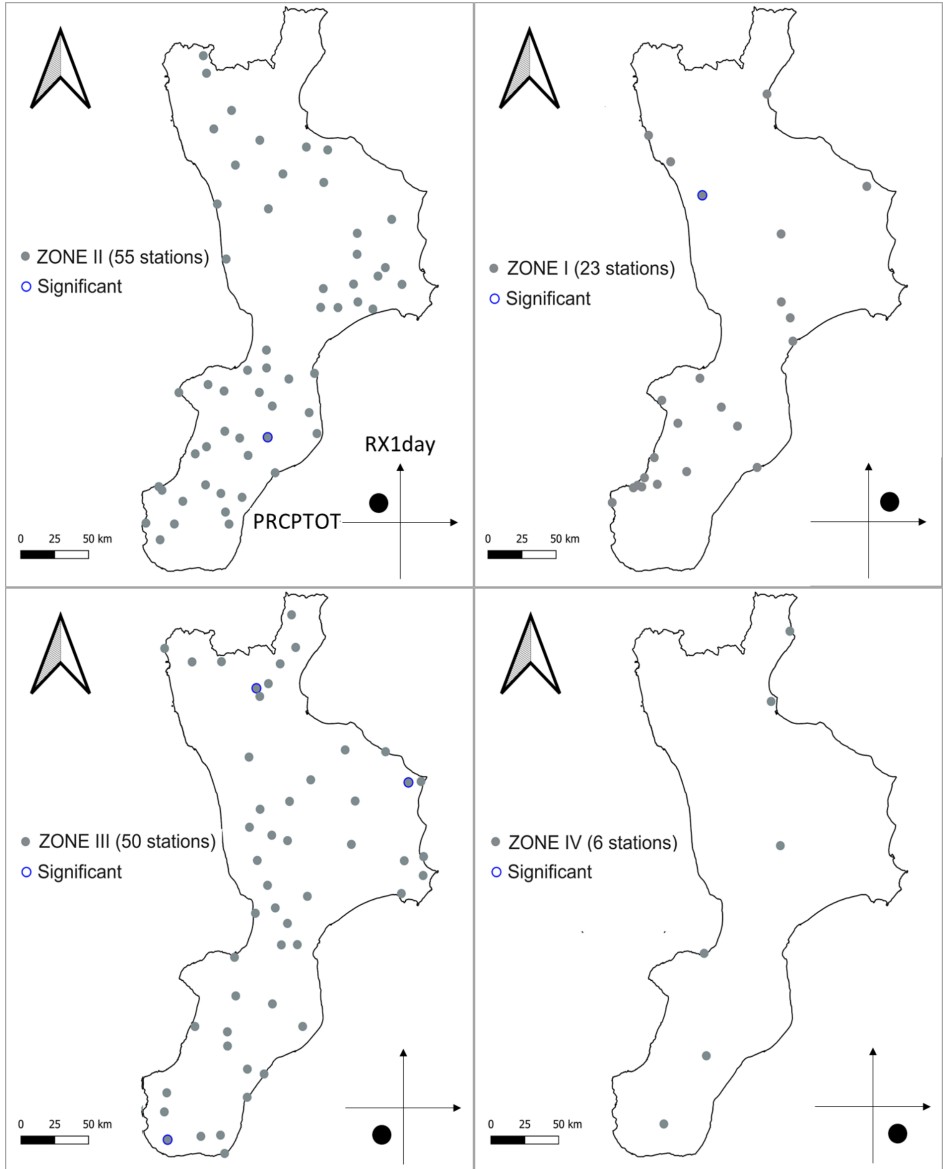

**Figure 5.** Four zones of compound trends of PRCPTOT and RX1day at the annual scale over Calabria from 1955 to 2023.

More generally, the analysis highlights the regional significance of the Calabrian peninsula in the northern Mediterranean basin, together with the availability of a dense and reliable monitoring network. These features are particularly helpful in supporting further analysis of climate evolution and related challenges in the study region. Specifically, the following sections will focus on the connection between extreme events in Calabria and SST since the latter is a critical variable in disentangling the seemingly counterintuitive outcome of increasing RX1day trends in a general context of decreasing PRCPTOT.





## 3.2 Observed and projected SST warming

Figure 6 shows the evolution of both the observed and projected yearly average SST in the Mediterranean basin. The increasing trend is almost monotonic for both observations and projections, which are in good agreement with each other. Concerning observations, substantially the same difference emerges when looking at the average values of the first and last five years of both ERA5 and CMEMS datasets, of about 1.2 °C. Such a difference, which agrees with previous literature (Mohamed et al., 2019; Pastor et al., 2020; Sannino et al., 2022), is approximately the same as that found in the period September-December, which will be further addressed with the WRF simulations.

Concerning projections, temperature changes were evaluated by comparing the future periods 2040-2069 and 2070-2098 with the reference period 1985-2014 for both SSP scenarios. At the annual scale, the projected temperature increase for the SSP3-7.0 scenario is averagely equal to 1.6±0.4 °C and 2.6±0.7 °C for the 2040-2069 and 2070-2098 period, respectively. Concerning the SSP5-8.5 scenario, instead, the differences are averagely equal, for the same periods, to 1.8±0.5 °C and 3.2±0.8 °C, respectively (not far from what was already projected by the CMIP5 models, e.g. Sannino et al. (2022)). Such as with observations, the differences detected at the annual scale are not far from those found in the period September-December, as highlighted by an analysis performed at the monthly scale (Figs. 7 and 8). The values found for this 4-month period are equal to 1.6±0.4 °C and 2.7±0.7 °C for SSP3 and 1.9±0.5 °C and 3.4±0.8 °C for SSP5, considering the 2040-2069 and 2070-2098 period, respectively. In particular, the 3$^{rd}$ quartile in the period 2070-2098 is equal to 3.0 °C with SSP3-7.0 and 3.9 °C with SSP5-8.5.

The period selected for the high-resolution weather simulations was September-December 2019. In that time interval, the average SST recorded was approximately 1.3 °C above the average SST of the first five years with available observations. Therefore, a homogeneous 1 °C reduction of the SST fields is a reasonable choice to trace back SST average conditions to the early '80s. On the other hand, the average SST conditions in September-December 2019 were approximately 0.9 °C higher than in the reference period 1985-2014. Therefore, a homogeneous increase of 3 °C of the SST fields allows reproducing the warmest projected conditions for the SSP5-8.5 scenario in the farthest future period 2070-2098. Hereafter, the simulation considering the actual SST conditions will be referred to as SST0, the past scenario with uniformly reduced SST values as SST-1, and the future scenario with uniformly increased SST values as SST+3.

## 3.3 High-resolution atmospheric simulations

The season most prone to extreme hydrometeorological events in southern Italy is the fall (from September to December) when cold-air intrusions reach the marine boundary layer with still high SST (Noyelle et al., 2019). For example, during the 2019 fall season, 20 precipitation events were recorded in southern Italy, among which 2 extreme events produced very intense precipitations (more than 200 mm of accumulated precipitation in some locations) and powerful wind speeds (gusts of up to 100 km/h), causing severe floods (Marsico and Rotundo, 2019; Fusto et al., 2019). The time sequence of the 20 events identified and labeled during the analyzed season is depicted in Fig. 9, in which the average daily precipitation values obtained for the Calabria region by spatial interpolation of the 150 available gauges are shown.




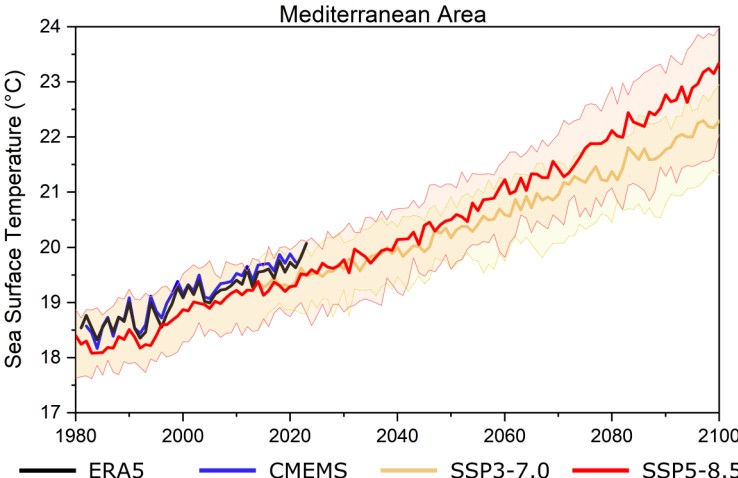

**Figure 6.** Yearly average SST (°C) in the Mediterranean Basin. For SSP3-7.0 and SSP5-8.5 scenarios, the thicker lines indicate the median for the model simulations, while the colored bands highlight the 10[th] and 90[th] percentiles, respectively.

Model validation with SST0 was carried out for all the 20 events detected. While Figures S1-S20 in the Supplementary show the observed and simulated accumulated precipitation maps for all the events, Figure 10 compares the verification indices FSS and CSI achieved for SST0, SST-1, and SST+3 scenarios, and the ERA5 reanalysis. Specifically, Figure 10a shows the FSS, emphasizing how the dynamic downscaling is crucial for an accurate representation and how it outperforms the lower-resolution ERA5 performance. Considering all the 20 events, the median FSS increases for larger window sizes as expected, exceeding 0.5 for the largest sizes, which typically indicates valuable skill (Roberts and Lean, 2008; Necker et al., 2024). Furthermore, it is noteworthy that increasing or decreasing the SST values degrades performance, highlighting the extent to which SST representation affects the simulations. Due to its higher difference from observed SST, SST+3 performance does not increase as much as SST-1 with increasing window size. Still, the SST+3 simulation remains generally better than the reanalysis. Considering the CSI index (Fig. 10b), all models perform relatively well for smaller precipitation thresholds. In contrast, as the threshold increases, the simulation with SST0 performs the best. It can also be seen that ERA5 has a median of 0 when considering the threshold of 30 mm, indicating that at least 50% of the events have no prediction skill.

After validating the simulations against observational data, we conducted an in-depth analysis of how varying sea surface temperature boundary conditions influence precipitation patterns. Considering the whole innermost domain at convection-permitting resolution (D02 in Fig. 1b), precipitation increases as the SST increases. In fact, the simulated accumulated precipitations for the 20 events analyzed are always lower for the past scenario (SST-1) and higher for the future scenario (SST+3) than the actual conditions (SST0) (Table A1 in the Appendix), indicating higher precipitation when a more significant amount of water vapor is available in the atmosphere due to a more significant contribution of evapotranspiration from a warmer sea. Specifically, the precipitation amounts (spatial average) collected for all events with different SSTs are quite well linearly cor-





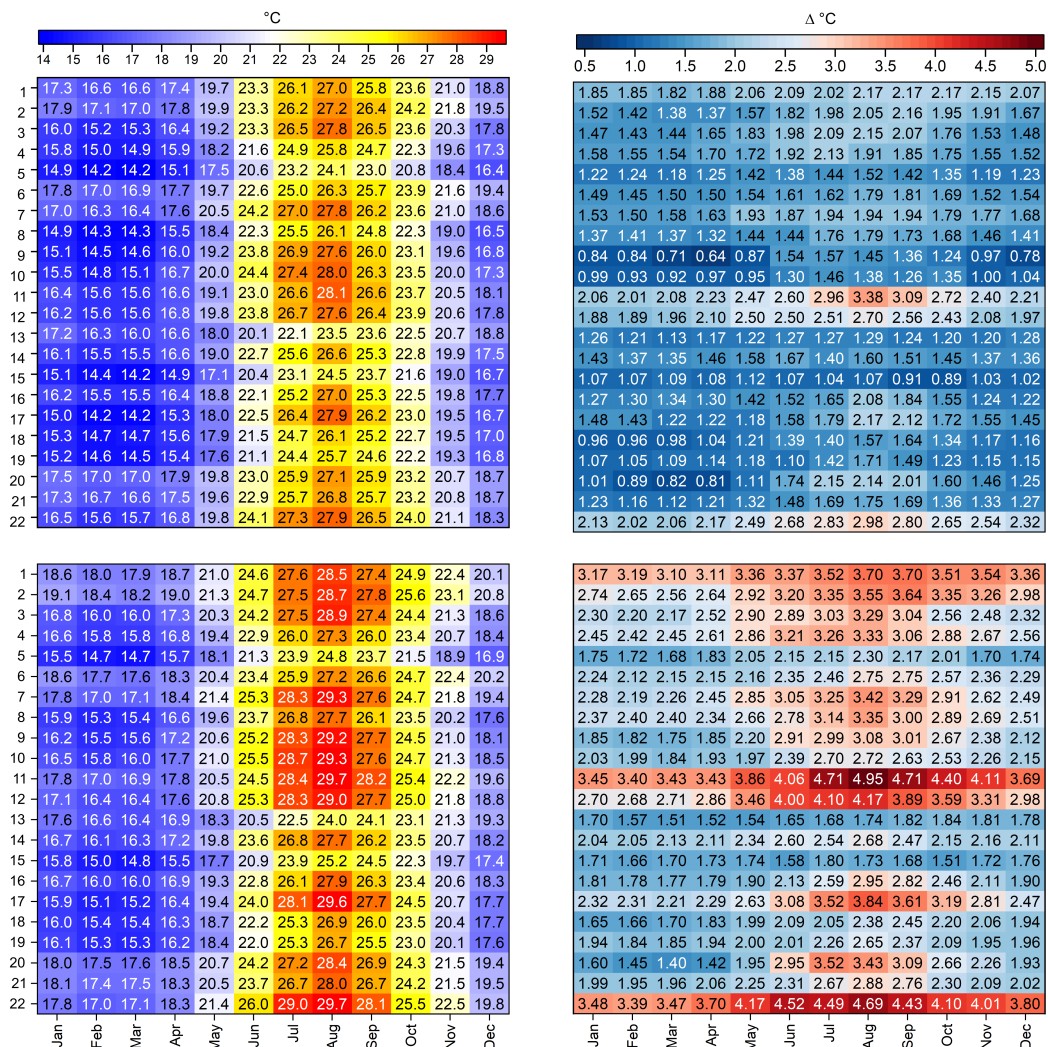

**Figure 7.** Monthly average SST (°C) simulated by the 22 GCMs with the SSP3-7.0 scenario. The tables in the first column indicate the absolute values, while those in the second column indicate the increase compared to the historical simulations from 1985 to 2014. The tables in the first row refer to the period 2040-2069, and those in the second row to the period 2070-2098.

related so that it is $P_{SST-1} = 0.92 \cdot P_{SST0} - 1.4$ ($r = 0.99$) and $P_{SST+3} = 1.14 \cdot P_{SST0} + 6.4$ ($r = 0.95$), with $P_{SSTx}$ [mm] indicating the accumulated precipitation with the given SST boundary condition.

If only accumulated rainfall on land is considered rather than over the whole domain D02 (Table S1), the linear correlations between $P_{SST0}$ and $P_{SST-1}$ and $P_{SST+3}$ still hold, being $P_{SST-1} = 0.94 \cdot P_{SST0} - 1.1$ ($r = 0.99$) and $P_{SST+3} = 0.88 \cdot P_{SST0} + 8.5$ ($r = 0.94$), respectively. Nevertheless, the increase in precipitation in response to SST increase is no longer always respected, especially for higher precipitation, as the slopes of the linear equations (higher for $P_{SST-1}$ in this case)



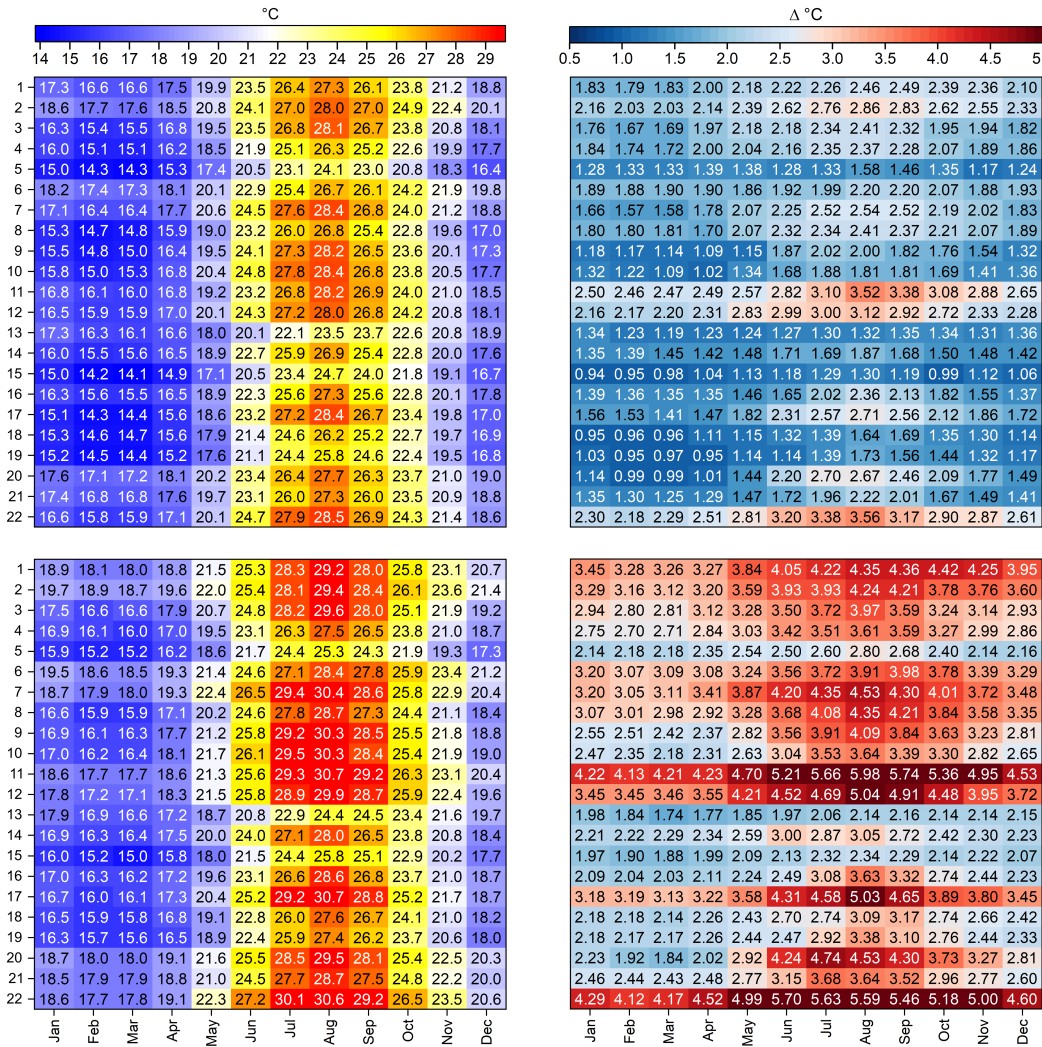

**Figure 8.** Monthly average SST (℃) simulated by the 22 GCMs with the SSP5-8.5 scenario. The tables in the first column indicate the absolute values, while those in the second column indicate the increase compared to the historical simulations from 1985 to 2014. The tables in the first row refer to the period 2040-2069, and those in the second row to the period 2070-2098.

suggest. Fig. 11 sets out the percentage of difference in precipitation-on-land between simulations using modified boundary conditions and the SST0 boundary condition compared to the accumulated precipitation simulated with SST0. Opposite trends arise considering SST-1 and SST+3. In the case of low SST0 precipitation values (i.e., light events), colder SST conditions lead to markedly reduced rainfall (approximately from -25% to -50%). In comparison, warmer SST conditions produce the opposite effect, with increases up to +75% and even more. However, when accumulated precipitation values with SST0 increase (i.e., in the case of heavier events), both the colder/warmer SST-induced precipitation differences tend to cancel out. For the two



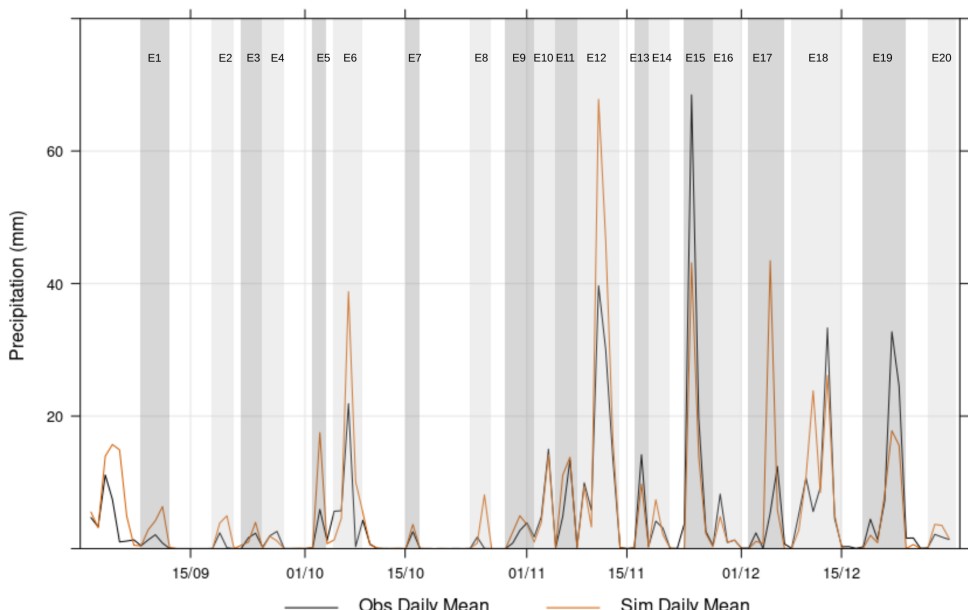

**Figure 9.** Temporal evolution of daily mean precipitation [mm] for the Calabria region in the analyzed fall 2019 period. The black line indicates the observed values obtained by spatial interpolation of observations, and the orange line represents the results of the SST0 simulation. In the figure, each event is identified and labeled.

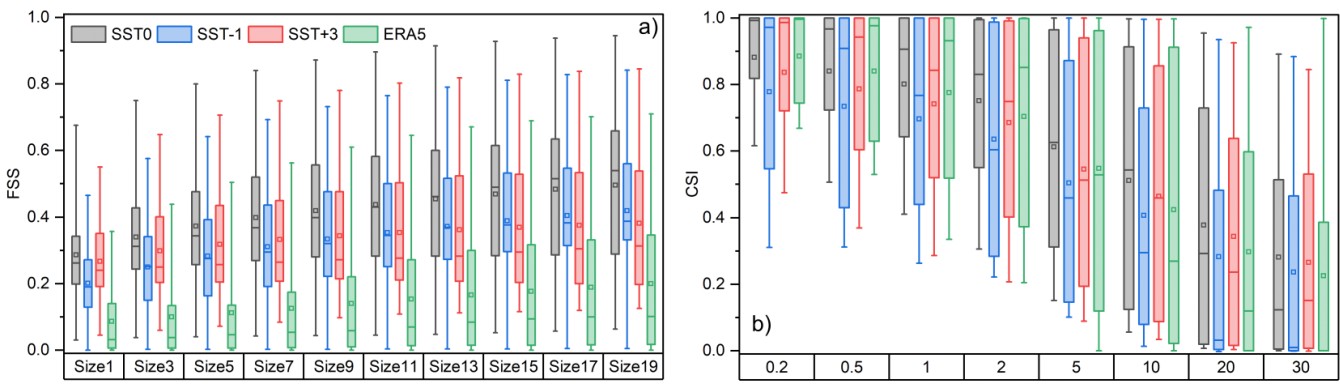

**Figure 10.** Box and whisker plots of the Fractional Skill Score (a) and the Critical Success Index (b) obtained by considering all the identified events and the three SST scenarios over the Calabria region. In the boxplots, the bottom and up whiskers indicate the 5th and 95th percentile, respectively; the box limits the 1st and the 3rd quartile, respectively, and the small line in the box is the median. The squares represent the average values.





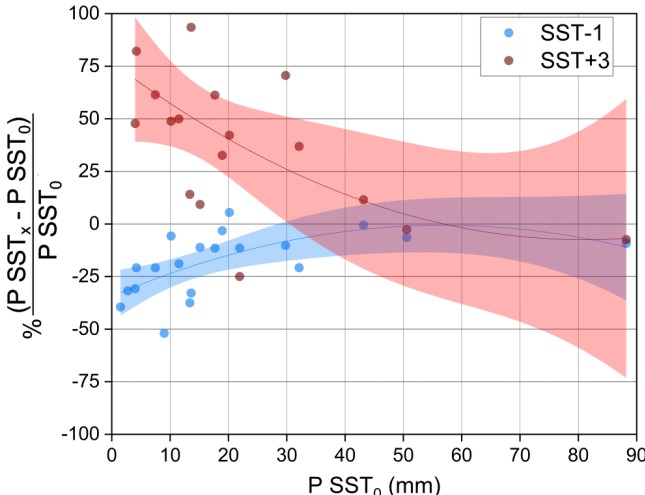

**Figure 11.** Percentage variation of the accumulated (spatially-averaged) overland precipitation for each event with SST-1 (blue dots) and SST+3 (red dots) boundary conditions vs. accumulated overland precipitation with SST0 boundary conditions.

heavier events, the precipitation differences for SST-1 are equal to -6% and -9%, while for SST+3, they are equal to -3% and
310    -7%, respectively.

The counterintuitive behavior of overland accumulated rainfall, which, for the heavier events, tends to decrease despite the higher SST values, is linked to the changes in the spatial pattern of the precipitation fields in the domain and, in particular, to the eastward shift of the rainfall peaks. Figure 12 depicts with circles the locations of the barycenters of the precipitation patterns calculated as described in section 2.4. In the figure, the circles related to the same event but with different SST boundary
315    conditions are connected by blue (SST-1 to SST0 barycenters) and red (SST0 to SST+3 barycenters) lines. For each triplet, the size of the red circle, representing the average precipitation of the pixels exceeding the 95[th] percentile for that event, is always bigger than the sizes of the blue (SST-1) and grey (SST0) circles. Indeed, such as for average precipitation, also rainfall peaks (Table S1) are linearly correlated ($P_{SST-1} = 0.88 \cdot P_{SST0} - 1.6$ with $r = 0.98$ and $P_{SST+3} = 1.13 \cdot P_{SST0} + 28.6$ with $r = 0.92$, respectively). Besides the amount of rainfall peaks, however, for our purposes, their different locations are particularly
320    important. Figure 12 highlights that in most of the cases, the SST+3 events are shifted eastwards of the respective SST0 events. Some of them, among which the rainiest events no. 6 and 12, are relocated over the Ionian Sea, saving the land from the heaviest rainfall. Such an effect was found especially for the cyclonic circulations that cross the Ionian Sea before reaching the eastern Calabrian coast, which are, in general, the heaviest and most impacting. The warmer boundary conditions modify appreciably the circulation pattern (for example, for event no. 12, the blue circle is over the Tyrrhenian Sea and the red one
325    over the Ionian Sea) and, mainly, foster the upload of significant moisture content from the sea, enhancing instability so that the storms explode over the sea before reaching the coastline.





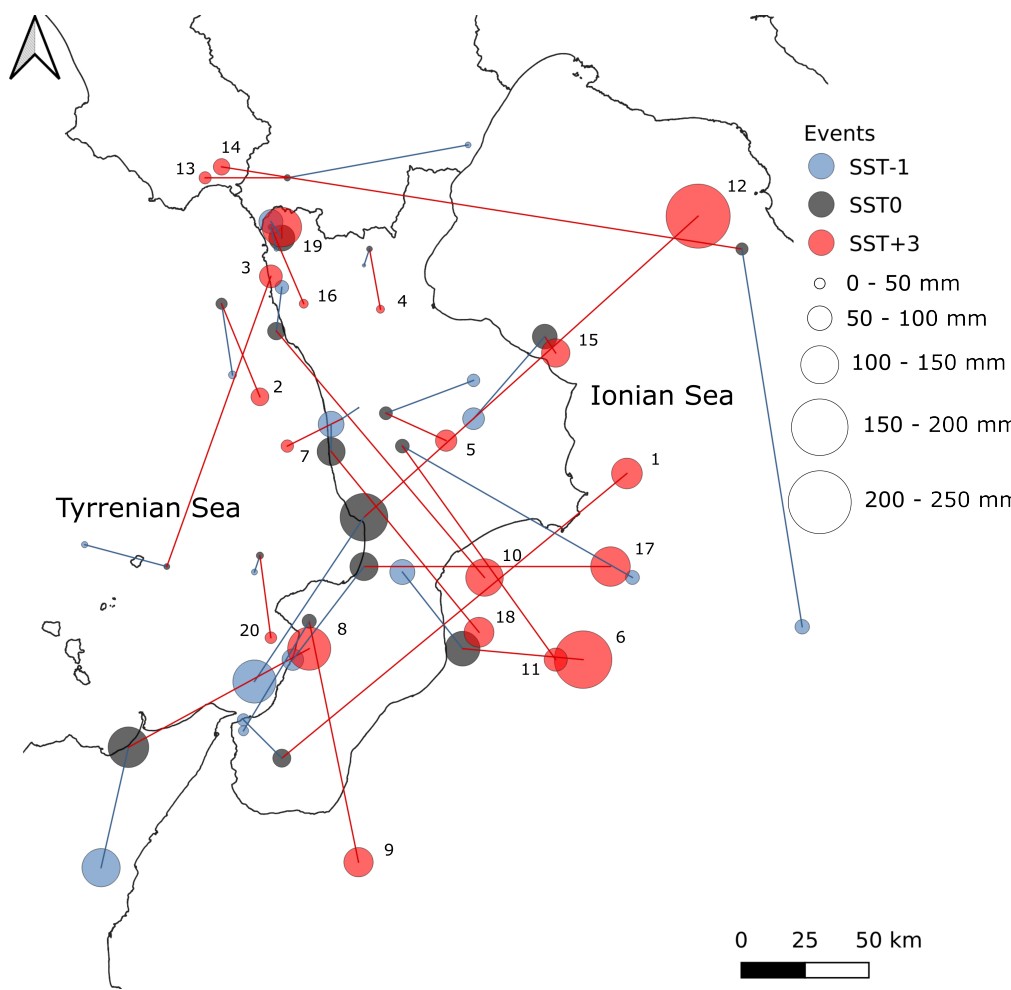

**Figure 12.** Locations of the centers of mass of the accumulated precipitation of the 20 analyzed events, considering the innermost domain, with SST-1 (blue filled circles), SST0 (dark grey), and SST+3 (red).

Event no. 12 was analyzed in detail to understand better the dynamics leading to the eastward shift of the precipitation pattern. Figure 13 shows the average convective available potential energy (CAPE) during the night preceding the most intense rainfall occurring during that event (Figs. 13a, c, and e), together with the average water vapor mixing ratio, vertical wind and vertical velocity omega profiles (Figs. 13b, d, and f) for a specific cross-section (red line in Fig. 13a). The experiment produced a substantial increase of CAPE on the Ionian Sea as SST increased, indicating a more considerable offshore instability for this kind of event in future scenarios. The vertical profile of omega, water vapor mixing ratio, and vertical wind at the selected cross-section confirmed this result. Negative values of omega, related to vorticity advection in the atmosphere (e.g., Lenderink et al. (2017)), describe unstable conditions, which were shifted eastwards with increasing SST. Also, the specific humidity developed in the atmosphere got higher over the Ionian Sea, and ascending currents were stronger. Therefore, Fig. 13 underscores that





**Figure 13.** Left: Average CAPE during the 8 hours preceding the most intense rainfall during event no. 12: a) SST-1, c) SST0, e) SST+3. Right: vertical profile of omega (color bar), water vapor mixing ratio (isolines), and vertical wind (upward or downward arrows for direction; speed proportional to the reference vector) for b) SST-1, d) SST0, f) SST+3.



the atmospheric instability and the moisture content can be shifted eastwards and offshore while moving towards warmer SST conditions.

### 3.4 Comparison with previous studies, limitations and insights

The experiments performed are part of a research line aimed at disentangling the expected influence of sea surface warming, even combined with orography, on cyclonic events features in Southern Europe (e.g., Miglietta et al. 2011; Meredith et al. 2015; Pytharoulis 2018; Noyelle et al. 2019; Varlas et al. 2023; Ricchi et al. 2023). In addition, for the first time, to the best of the authors' knowledge, they analyze an entire season with 20 events at a convection-permitting resolution, highlighting variations in features like average and peak intensity and tracking. Of course, to achieve a more comprehensive overview of future conditions, the focus on SST should be complemented by other information. For example, contrary to the pseudo-global warming approach used here, generally, the atmosphere's vertical profile is also expected to be modified. Modifying only SST can lead to an overestimate in the vertical gradient of temperature and, therefore, biased effects on precipitation. In this case study, however, the events are analyzed individually and can be considered as isolated, such as, e.g., the cold air intrusions that typically occur in this area at this time of year, and are therefore plausible also in a global warming scenario, not undermining the relevance of our experiments. Besides local changes in atmospheric properties, the expected frequency of cyclones, given the large-scale circulation dynamics changes induced by climate warming, is another relevant piece of information missing. However, recent literature does not provide a unanimous response regarding the projection of heavy rainfall events in the central Mediterranean.

Analyzing the decomposed contribution from atmospheric thermodynamics and dynamics in CMIP5 projections, Pfahl et al. (2017) concluded that reduced cyclone frequency is expected in the broader Mediterranean region. Considering a higher resolution provided by the Med-CORDEX (CMIP5-derived) ensemble, Reale et al. (2022) for the end of this century found that the overall frequency decrease and weakening of cyclones moving across the Mediterranean will be compensated, in the central part, by increased wind speed and precipitation rate. Using the Euro-CORDEX ensemble, Matte et al. (2022) projected for the Central Mediterranean an increase in the number of events exceeding the 90[th] percentile for the larger-sized precipitation systems and a small decrease for medium-sized systems. Also, Hosseinzadehtalaei et al. (2020), projecting intensity–duration–frequency curves over Europe, showed that the sub-daily extreme precipitation events with the highest return periods are expected to become more frequent. Another analysis based on the Euro-CORDEX ensemble and focused on precipitation overland (Tramblay and Somot, 2018) emphasizes an overall increase in 20-year extreme precipitation in the northern Mediterranean coast and a decrease in the southern coast, in agreement with the ongoing trends described in this paper (section 3.1). Finally, by further increasing the spatial resolution with an ensemble of convection-permitting regional climate models under the RCP8.5 forcing scenario, Müller et al. (2024) found that more intense, heavy, and severe precipitation events must be expected in southern Italy, especially during fall in the form of landfalling and geographically forced events. Moving to CMIP6 projections, Fernández-Alvarez et al. (2023), analyzing the Community Earth System Model Version 2 (CESM2; Danabasoglu (2023)) under the SSP5-8.5 scenario, recognized a contrasting contribution of moisture transport from the Mediterranean Sea and the North Atlantic Ocean to the western continental area of the Mediterranean basin, with the former increasing precipita-





tion and the latter decreasing it. Finally, Anav et al. (2024) performed a dynamical downscaling of the MPI-ESM1-2-HR model (model no. 18 in Table 1). Concerning precipitation, they only provided results at the seasonal scale, revealing a substantial reduction. However, they demonstrated that the high-resolution air-sea coupling improved the representation of high-impact events like marine heat waves.

In the overview of heavy rainfall projections summarized above, the risk of high-impact events increases, in general, with 375 increasing resolution of the simulation. This outcome can be explained by considering the more relevant effect of higher-resolution complex topography in triggering convection (Ricchi et al., 2023). Using very high-resolution (2 km) modeling, our study demonstrated that the interaction of an increasingly warmer and humid atmosphere with coastal orography, which impacts the development of convective systems even offshore (Khodayar et al., 2021), further enhances the moisture loading and favors their maturity so that rainfall peaks occur before landfall, especially for more unstable systems. In this way, given a 380 precipitation rate increase in the overall domain (i.e., considering both land and sea), the overland precipitation rate increases for small to medium-sized events, making them more dangerous and impacting, but is constant or even reduces slightly for heavy events, which, however, remain still dangerous. Therefore, the overall tendency can be summarized as an expected increasing frequency of impacting overland events rather than increasing intensity.

## 4   Conclusions

Understanding how heavy precipitation events evolve in the Mediterranean basin under a changing climate is an increasingly compelling research topic with countless practical implications. In just the past few years, the region has experienced several extreme events with severe consequences, including Storm Daniel in Libya in 2023 (Normand and Heggy, 2024), the Valencia (Spain) event in 2024 (Amiri et al., 2025), and two consecutive events in Emilia Romagna (Italy) in 2023 and 2024 (Arrighi and Domeneghetti, 2024; Ferrari et al., 2025), all of which caused significant damage and fatalities. Along with these storms, 390 many other intense but less destructive events affected this climate change 'hotspot'.

The research provides a comprehensive investigation of the topic. First, a combined trend analysis of annual and maximum one-day precipitation over the EURO-CORDEX domain revealed divergent trends along much of Southern Europe, with heavy precipitation increasing but total annual precipitation decreasing. These trends were compared with those achieved by ground-based measurements in the Calabrian peninsula, leveraging an observational dataset spanning nearly 70 years. The results 395 confirmed the reliability of the ERA5-Land dataset and pointed out that this subregion can be considered representative of much of the northern Mediterranean coast.

Next, we examined the role of sea-atmosphere-orography interactions in explaining heavy precipitation enhancement despite the overall drying trend. In particular, we isolated the impact of sea surface warming by simulating an especially intense rainy season in the Calabrian peninsula and comparing current SST conditions with both past (SST-1) and future (SST+3) scenarios. 400 The convection-permitting resolution of the dynamical downscaling approach allowed for a highly detailed reconstruction of the different cyclone tracks and the resulting precipitation patterns induced by varying SST boundary conditions. The numerical experiments explained the enhancement of overland heavy precipitation events' frequency but did not indicate an increase in



peak intensity since the most extreme events should tend to produce their highest rainfall totals over the sea before reaching land.

Beyond these findings, the study's main methodological contribution is demonstrating that only high-resolution, convection-permitting analyses can accurately capture key processes unique in orographically complex regions like the one addressed. The detailed approach adopted helps explain the seemingly contradictory trends of increasing daily maximum rainfall and decreasing annual total precipitation, a pattern that can be generalized to much of the northern Mediterranean coast. As hyper-resolution climate simulations become more widely available, they will allow for further validation and refinement of these

results.

Future research will build on this analysis by incorporating newly available convection-permitting climate simulations, which account for additional global warming-induced processes beyond SST increases. A key focus will be the hydrological impact, as the expected rise in heavy precipitation event frequency could disproportionately elevate flood risk. In fact, when soil is already saturated from a prior heavy rainfall event, subsequent storms occurring in quick succession can significantly amplify

flood impacts.

*Data availability.* The daily precipitation at the high resolution simulated by the WRF model in the innermost domain D02 can be downloaded at https://doi.org/10.5281/zenodo.14848874 for all three SST scenarios. Precipitation gauges data are available upon request from the Centro Funzionale Multirischi – ARPACAL (2025; http://www.cfd.calabria.it/index.php/dati-stazioni/dati-storici. ERA5-Land can be retrieved from https://cds.climate.copernicus.eu/datasets/reanalysis-era5-land. ERA5 Data can be retrieved from https://cds.climate.copernicus.

eu/datasets/reanalysis-era5-single-levels-monthly-means?tab=overview. CMIP6 Data can be retrieved from https://doi.org/10.20350/digitalCSIC/15492. CMEMS Data can be retrieved from https://doi.org/10.48670/moi-00169.

## Appendix A

Table A1 provides a detailed analysis of the 20 precipitation events identified in the September-December 2019 period. It quantifies average precipitation (mm) across the entire internal domain of the WRF simulation, overland-only precipitation,

and precipitation above the 95$^{th}$ percentile, simulated by SST0, SST-1, and SST+3 scenarios, respectively.




| ID | Start - End Date | n° days | P̄ D02 SST0 | P̄ D02 SST-1 | P̄ D02 SST+3 | P̄ Land SST0 | P̄ land SST-1 | P̄ land SST+3 | $\bar{P} > 95^{th}$ SST0 | $\bar{P} > 95^{th}$ SST-1 | $\bar{P} > 95^{th}$ SST+3 |
|---|---|---|---|---|---|---|---|---|---|---|---|
| 1 | 09/09-11/09 | 3 | 10.4 | 4.7 | 23.8 | 13.6 | 9.1 | 26.3 | 71.7 | 48.9 | 121.9 |
| 2 | 19/09-20/09 | 2 | 6.2 | 3.0 | 14.2 | 13.4 | 8.4 | 15.3 | 45.1 | 31.7 | 70.8 |
| 3 | 23/09-24/09 | 2 | 2.1 | 1.6 | 12.4 | 2.7 | 1.9 | 19.3 | 23.9 | 25.1 | 90.7 |
| 4 | 26/09-27/09 | 2 | 2.0 | 1.0 | 4.0 | 4.0 | 2.8 | 5.9 | 23.2 | 13.9 | 32.1 |
| 5 | 03/10-03/10 | 1 | 14.7 | 13.2 | 18.4 | 15.1 | 13.4 | 16.5 | 51.7 | 49.9 | 84.9 |
| 6 | 05/10-09/10 | 5 | 27.9 | 22.5 | 53.7 | 32.1 | 25.4 | 43.9 | 138.0 | 99.4 | 228.1 |
| 7 | 16/10-16/10 | 1 | 0.5 | 0.3 | 5.5 | 1.5 | 0.9 | 7.2 | 8.6 | 5.4 | 49.5 |
| 8 | 25/10-26/10 | 2 | 15.1 | 11.8 | 22.1 | 20.2 | 21.3 | 28.7 | 161.1 | 151.6 | 170.7 |
| 9 | 20/10-01/11 | 3 | 10.7 | 6.0 | 20.5 | 11.5 | 9.3 | 17.2 | 56.4 | 40.1 | 116.5 |
| 10 | 02/11-04/11 | 3 | 11.4 | 9.0 | 26.4 | 17.7 | 15.6 | 28.5 | 69.7 | 54.0 | 148.6 |
| 11 | 06/11-07/11 | 2 | 20.7 | 19.9 | 30.0 | 18.9 | 18.3 | 25.1 | 54.8 | 55.5 | 91.9 |
| 12 | 09/11-13/11 | 5 | 53.4 | 48.1 | 68.1 | 88.2 | 80.1 | 81.7 | 189.2 | 170.9 | 254.1 |
| 13 | 17/11-17/11 | 1 | 7.8 | 7.1 | 12.1 | 10.2 | 9.6 | 15.1 | 26.0 | 23.8 | 48.5 |
| 14 | 19/11-20/11 | 2 | 11.9 | 9.4 | 18.5 | 9.0 | 4.3 | 19.4 | 48.8 | 57.5 | 65.6 |
| 15 | 24/11-26/11 | 3 | 39.6 | 37.3 | 42.6 | 43.1 | 42.9 | 48.1 | 98.2 | 86.5 | 112.7 |
| 16 | 28/11-30/11 | 3 | 1.7 | 1.2 | 4.0 | 4.2 | 3.3 | 7.7 | 23.3 | 18.2 | 35.7 |
| 17 | 03/12-06/12 | 4 | 14.6 | 10.2 | 21.9 | 21.9 | 19.4 | 16.4 | 111.8 | 85.7 | 156.7 |
| 18 | 09/12-14/12 | 6 | 37.5 | 32.4 | 44.4 | 50.6 | 47.3 | 49.3 | 111.3 | 102.3 | 118.6 |
| 19 | 19/12-23/12 | 5 | 12.2 | 10.8 | 23.9 | 29.8 | 26.8 | 50.8 | 103.5 | 95.2 | 156.8 |
| 20 | 28/12-30/12 | 3 | 4.8 | 3.5 | 8.9 | 7.5 | 5.9 | 12.0 | 28.6 | 23.6 | 46.5 |

**Table A1.** Main features of the events identified in the analyzed period: event ID; starting and ending dates; duration in number of days; average precipitation $\bar{P}$ in the innermost D02 domain simulated by SST0, SST-1, and SST+3 scenarios, respectively; average overland precipitation $\bar{P}$ simulated by SST0, SST-1, and SST+3 scenarios, respectively; average of the accumulated precipitation $\bar{P}$ exceeding the $95^{th}$ percentile simulated by SST0, SST-1, and SST+3 scenarios, respectively. All values are expressed in mm.



*Author contributions.* Conceptualization: A.S., G.M.; Data curation: L.F., G.N., Formal analysis: all authors; Investigation: all authors; Methodology: L.F., G.N., J.C., with support of A.S. and G.M.; Software: L.F., G.N.; Supervision: G.M.; Visualization: L.F., G.N., Writing – original draft: A.S. with support of all authors; Writing – review & editing: all authors.

*Competing interests.* The authors declare that they have no conflict of interest.

*Acknowledgements.* This work was funded by the Next Generation EU - Italian NRRP, Mission 4, Component 2, Investment 1.5, call for the creation and strengthening of 'Innovation Ecosystems', building 'Territorial R&D Leaders' (Directorial Decree n. 2021/3277) - project Tech4You - Technologies for climate change adaptation and quality of life improvement, n. ECS0000009. This work reflects only the authors' views and opinions, neither the Ministry for University and Research nor the European Commission can be considered responsible for them. We thank the "Centro Funzionale Multirischi" of the Calabrian Regional Agency for the Protection of the Environment for providing the
observed precipitation data.



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
