# Peer review of "Increasing Daily Extreme and Declining Annual Precipitation in Southern Europe: A Modeling Study on the Effects of Mediterranean Warming"

_EGUsphere, 2025_

## Author Comment (AC1)

We would like to warmly thank the Referee for their thorough review of our paper. The comments and suggestions provided will contribute significantly to improving the quality of our paper, making it more effective and clearer. Please find below our point-by-point responses (in red text).

**General Comments**

This study investigates the evolution of heavy precipitation events in the Mediterranean basin under a changing climate, focusing on the role of sea-atmosphere-orography interactions. Using both observational data and numerical simulations, the authors assess the trends in annual and maximum precipitation, with an emphasis on the Calabrian peninsula. They also explore how sea surface temperature changes might impact precipitation patterns, particularly during intense rainy seasons. The study emphasizes the importance of high-resolution, convection-permitting models to capture key processes. While the topic is relevant and the approach is promising, several aspects of the manuscript require clarification and improvement to strengthen the overall impact and scientific contribution.

We thank the Referee for the positive feedback and for highlighting the most important aspects of the research. We acknowledge that various aspects of the analysis require further strengthening. Below, we attempt to address the suggested concerns and comments.

**Introduction**

This section is overall well-written and provides relevant background. However, it does not clearly address the specific research gaps this study aims to fill. The authors should explicitly state the study's novelty (for example, the use of 20 real-case events at convection-permitting scale with calibrated SST perturbations) and how it builds on existing work.

The main aim of this paper is to investigate the extent to which Mediterranean Sea warming contributes to the seemingly counterintuitive increase in daily precipitation extremes in southern Europe, despite a general decline in annual precipitation. We agree with the Referee (and will modify the paper accordingly) that our aim to fill this research gap should be more clearly highlighted in the Introduction, along with the main strengths of the analysis performed (mainly, the convection-permitting resolution adopted in 20 real-world events with modified SST conditions, which allowed us to analyze in great detail the effect of sea-air-orography interactions on high precipitation features, and the special focus on overland precipitation, differently from the majority of the studies focusing on precipitation features over the whole domain).

**Data and Methods**

1. This section would benefit from greater clarity and conciseness. The SST perturbation approach is not clearly explained, details on the magnitude, spatial pattern, and implementation (e.g., uniform or spatially varying changes) are missing.

In the Data and Methods section, we presented only the datasets used for the SST analysis (LL114-131), including the extent of the area analyzed (L120). Then, in the Results section, we provided more details about the perturbation approach (LL262-269), including the

magnitude (from -1 °C to +3 °C compared to current conditions) and the spatial pattern and implementation (e.g., we claimed a homogeneous change). We agree that the SST perturbation approach can be presented in a clearer and more organic way in the Data and Section method, as pointed out also by the other Referees (e.g., Referee #3 comment no. 6).

2. The quadrant classification based on PRCPTOT and RX1day trends is also unclear, a clearer definition or simple diagram would help.

The quadrant classification is a method for highlighting the combined trends of two different variables. We will strive to explain it more clearly, even with a simple diagram like the one below. However, the representation of the results (Figs. 3 and 5) will be modified according to the comments of other referees (Referees #2, specific comments).

[Figure]

3. Additionally, the event identification method (starting the event the day before precipitation begins) is unconventional, as most studies define event onset based on a precipitation threshold or objective criteria, further justification is needed.

We sincerely thank the Referee for this comment, which points out a wrong statement in the manuscript, 'survived' from previous versions (it was related to some spin-up issues). Indeed, event selection was made objectively. Specifically, we used a fixed threshold of 0.8 mm on the average daily precipitation values simulated by the WRF model (SST0 simulation) over the Calabria region. Indirect confirmation of the use of this approach is given by Figure 9, from which it is clear that the day-before-precipitation approach was not used. We opted for this choice, rather than detecting the event based on the observations, because (a) as shown in Figure 9, the SST0 simulation performs very well in terms of timing, and (b) this choice makes it easier to further compare with the SST+3 and SST-1 simulations.

**Results and Discussion**
Overall, the results and discussion sections offer valuable insights, but they lack clear physical explanations to support the findings.

1. In the Trend Analysis section, the authors present an analysis of observed rainfall trends in Calabria but fail to adequately link these results to the region's orographic features. While the orography is mentioned multiple times, there is no attempt to explain how it might be influencing the observed rainfall patterns, particularly in terms of total maximum rainfall. This lack of connection makes it harder to understand why different trends are observed.

While several studies are available concerning features of precipitation spatial patterns in Calabria, which will be duly cited, a new analysis will be proposed and discussed based on the available dataset, in which geographical and orographic features like latitude (N), longitude (E), elevation, slope, aspect, and distance from the sea are analyzed to study how effectively they affect observed average annual precipitation (Precp_avg), as well as trends of PRCPTOT and RX1day in terms of Sen's slope. By applying the k-means method within the framework of Principal Component Analysis (PCA), the area of interest is preliminarily divided into four homogeneous clusters, as shown in the figure below.

[Figure]

The findings achieved so far highlight that orography is more closely associated with the spatial pattern of observed precipitation than with the temporal trends of PRCPTOT and RX1day.

2. In the Observed and Projected SST Warming section, the authors choose SSP changes of -1 and +3°C for their simulations, but the rationale behind these choices is unclear. Were these values based on region-specific data, or were they generalized from the broader Mediterranean basin? It would also be more logical for the authors to focus more on their study area (Domain 3) rather than a larger region.

We believe that the rationale for the choices is explained in this section. However, as also noted in our reply to the previous Data and Methods comment no. 1, we'll strive to present the entire approach in a more straightforward and organic manner. The spatial extent of the analyzed region corresponds to the IPCC Atlas Mediterranean region (L120). However, we agree that, to avoid confusion among different domains and study areas, it is more straightforward to focus on the external domain (D01) of the WRF simulation, so we changed Figure 6 and replaced Figures 7 and 8 with a new one, following also suggestions provided by Referee #2 (comment no. 5). The new figures are shown below (in the spaghetti graphs, the dotted line represents the median behavior). We observed a slight increase in projected SST. Further details will be provided in the revised text.

[Figure]

3. For the WRF Simulation Evaluation, the authors chose to spatially interpolate the observations. However, it's unclear whether this is the most appropriate method for comparison. Interpolating observations can introduce uncertainty and might not accurately

reflect the spatial variability of the actual observations. It would be useful for the authors to justify why they chose this method.

We thank the reviewer for the comment. Spatial interpolation was performed using an Inverse Distance Weighting (IDW) technique (i.e., an exact interpolator), as specified at L111-113 in the original manuscript, with a total of 150 stations (134 "historical" and 16 "recent" gauges), resulting in a density of approximately 1 station per 100 km$^2$. While we acknowledge that using an interpolation method (which inherently means adding another model) adds uncertainty, we observe that: (a) a gauge is not necessarily entirely representative of the calculation cell of the atmospheric model in which it falls, especially in the numerous steep areas of the region, therefore also gauge-to-pixel direct comparison can be affected by some bias; (b) on the other hand, the density of the monitoring network in the region is high, preventing significant misinterpretations of the spatial patterns of precipitation; and (c) while literature is plenty of studies based on gridded datasets of questionable quality (the topic is partially addressed also in this paper, but for southern Italy please refer, e.g., to Cammalleri et al., 2024), in this case we have a fully validated spatially dense observational dataset which is a perfect candidate for the development of a reliable gridded dataset. We will add these comments to the text to justify our choice.

4. The Figure 12 showing the eastward shift of extreme rainfall events is visually appealing, but the authors do not explain the underlying mechanisms driving this shift. While they focus on a single event, there is no physical explanation for the observed trend, which limits the depth of the physical understanding gained from the analysis.

In the manuscript, we provide details of the underlying mechanisms of the shift through the example of event no. 12 (LL 327-337 and Fig. 13). However, considering also comments from other Referees (Referee #2 comment no. 7, and Referee #4 comment no. 6), we acknowledge that the explanation should be more general and straightforward and we will strive to provide more details in the revised version of the manuscript.

5. In the Comparison with Previous Studies section, the authors compare their findings to studies that account for global warming, rather than focusing only on SST. This comparison could lead to misleading conclusions because the underlying drivers of rainfall changes could differ between global warming and SST warming alone. More clarity on the boundaries of their analysis and comparison with relevant studies focusing on SST would enhance the discussion.

We thank the Referee for this comment. This Section will be expanded, providing more room for studies based solely on SST warming, to which we previously only hinted (LL340-341), adding others (e.g., Lin et al., 2023; Dutheil et al., 2022), and highlighting the boundaries of our analysis even more effectively. However, it is also of interest to us to refer to "complete" global warming scenarios to provide the whole picture, also because we are currently working on convection-permitting climate simulations in the study area (https://doi.org/10.5194/egusphere-egu25-15936), and they represent an area of future investigation for us (L411).

6. Lastly, Figures 7 and 8 present an overwhelming amount of information, making them difficult to interpret. The authors might consider finding an alternative way to present these results, perhaps by simplifying the figures or breaking them down into more digestible parts.

We agree with the Referee. This concern was also raised by other referees (Referee #2, comment no.5, Referee #3, comment no.5, and, partially, Referee #4, minor comment no. 15). Following Referee #2's suggestion, the representation of projected SST increase will be changed entirely, using only one multipanel figure with four spaghetti graphs, showing the SST increase compared to 1985-2014 in the periods 2040-2069 and 2070-2098, under the SSP3-7.0 and SSP5-8.5 scenarios, respectively. Additionally, according to comment no. 2 concerning the Results and Discussion section, the area on which we will base our calculations will no longer be the entire Mediterranean basin, but the external domain D01 (please refer to our reply to comment no. 2).

**Conclusions**

1. The conclusions section lacks impact and doesn't clearly tie the study's findings together. It doesn't explain how the research advances our current understanding. While the authors summarize their results, they could better highlight the practical implications of their work. For example, they could link their findings to how the study might help predict or mitigate future storms. It would also be useful to mention the key implications of these trends for climate adaptation or urban planning, especially in terms of how extreme precipitation affects flood risks.

We thank the Referee for their several valuable suggestions, which will be given due consideration. In particular, we will highlight the advancements in current understanding made possible by our high-resolution simulations and our focus on changes in overland precipitation, as well as how our research can contribute to mitigating flood risk. Indeed, this aspect was already hinted at in the last paragraph of the manuscript (LL412-415), but we will elaborate on it further.

2. Lastly, the statement that "only high-resolution, convection-permitting analyses can accurately capture key processes" is too strong and could benefit from further context.

We agree that this sentence is sharp and will strive to contextualize it better, while (anyway) avoiding making the concluding paragraph excessively long.

**References**

Cammalleri, C., Sarwar, A. N., Avino, A., Nikravesh, G., Bonaccorso, B., Mendicino, G., Senatore, A., and Manfreda, S. (2024). Testing trends in gridded rainfall datasets at relevant 475 hydrological scales: A comparative study with regional ground observations in Southern Italy, Journal of Hydrology: Regional Studies, 55, 101950.

Dutheil, C., M. Lengaigne, J. Vialard, S. Jullien, and C. Menkes (2022). Western and Central Tropical Pacific Rainfall Response to Climate Change: Sensitivity to Projected Sea Surface Temperature Patterns. *J. Climate*, **35**, 6175–6189.

Lin, K.-J., Yang, S.-C., & Chen, S. S. (2023). Sensitivity of extreme rainfall in Taiwan to SST over the South China Sea through modulation of marine boundary layer jet: A mei-yu front event during 1–4 June 2017. *Geophysical Research Letters*, 50, e2023GL104441.

---

## Author Comment (AC2)

We would like to warmly thank the Referee for their thorough review of our paper. The comments and suggestions provided, some of which particularly sound from a technical point of view, will contribute significantly to improving the quality of our paper, making it more effective and clearer. Please find below our point-by-point responses (in red text).

In their study "Increasing Daily Extreme and Declining Annual Precipitation in Southern Europe: A Modeling Study on the Effects of Mediterranean Warming" the authors investigate how projected changes in SST would unfold with respect to precipitation extremes of the Mediterranean north shore and in particular for the region of Calabria. In addition they elaborate on the hypothesis that there is a general trend of decreasing total annual precipitation along with increasing daily maximums, based on an analysis of reanalysis data and local observations. The paper addresses an important topic and the rationale is reasonable. The analysis is sound and well structured. Some arguments and conclusions may require a deeper analysis and discussion than currently presented, also given the limited significance of the trend analysis.

We thank the referee for the positive feedback and acknowledge that some aspects of the analysis require further strengthening. Below, we attempt to address the suggested concerns and comments.

1. The analysis of the ERA5-Land data for the EURO-CORDEX area does show negative trends but they are only significant for the Iberian peninsula and some regions of northern Africa. The findings for RX1day are even weaker. This lack of robustness should be addressed in more detail in the discussion. From ERA5-Land it looks like for the Calabria region, most pixels are seen in zone I of Fig. 3 and only a few in zone II which would contradict your statement about the match of ERA5-Land and the local observations. If you skipped the non significant values in Fig. 5 only a few would remain. What's the reason for the non-significance? Are there recurrent outliers in the observations?

The issue of statistically non-significant trends will be addressed more carefully in the discussion, in which we will also expand our references to literature addressing the same topic, e.g., the very recent work by Beranová et al. (2025).
Regarding the behavior of ERA5-Land in the Calabria region, it is useful, as suggested by other referees (specifically, Referee #2, major comment no. 4), to provide a data comparison between the gridded dataset and observations. The figure below illustrates the compound annual trends of PRCPTOT and RX1day over Calabria from 1955 to 2023 derived from both gauge-based and ERA5-Land datasets. It reveals that ERA5-Land mainly represents zones I and II, whereas ground truth measurements indicate that zones II and III are the majority. Notably, the statement of a possible match and potential alignment between ERA5-Land and the local observations is more supported in zone II (please refer to our reply to Referee #2, major comment no. 4, for a comparison of the PRCPTOT and RX1day trends separately).

[Figure]

2. For the simulations with the regional atmospheric model it is assumed that just the SSTs are changing according to certain SSP scenarios. The atmospheric properties remain unchanged which creates an inconsistency for the described future conditions. In your WRF configuration, the GHG settings seem to be constant across your simulations. Later WRF versions, e.g., with the CAM radiation scheme allow for an adjustment of GHG concentrations and also support different SSPs. With these updated atmospheric settings, you should obtain a more realistic interplay between sea surface and the atmospheric boundary layer, mostly due to radiation effects.

The main aim of our paper is to investigate the extent to which Mediterranean Sea warming contributes to the seemingly counterintuitive increase in daily precipitation extremes in southern Europe, despite a general decline in annual precipitation. Our scientific objectives lie in a robust "research line aimed at disentangling the expected influence of sea surface warming, even combined with orography, on cyclonic events features in Southern Europe" (L339-340). Therefore, adjusting GHG concentrations according to specific or more SSPs goes beyond our scope. Incidentally, making this choice would inherently imply the prior choice of a particular scenario and time period, which is not our intention. On the other hand, we are currently conducting convection-permitting climate simulations in the study area (https://doi.org/10.5194/egusphere-egu25-15936), which represent an area of future investigation for us (L411).

3. Another concern is the influence of large scale dynamics and patterns. From the text it is not clear whether you applied the updated SSTs to both domains. I assume you did so also for the outer domain. Adding this much energy to the full extent of domain one could considerably change the larger scale dynamics, patterns and feedback. Therefore, to corroborate your findings of translocated precipitation events, I recommend to create another

set of simulations to apply some spectral nudging to the outer domain, at least for the geopotential, to ensure consistency of the large scale structures also with respect to ERA5.

We uniformly applied temperature changes (+3 °C and -1 °C) to the SST fields on both calculation domains, including the outer domain. This aspect will be better specified in the revised version of the manuscript, as required also by other referees.
Concerning spectral nudging, following the Referee's suggestions, we performed two new experiments, both applying spectral nudging to the outer domain (D01) on the geopotential (ERA5 source) at heights above 500 m, for SST0 and SST+3 scenarios. The figure below shows the results of event 12: a) SST0 without nudging; b) SST0 with nudging; c) SST+3 without nudging; d) SST+3 with nudging. While precipitation fields undergo slight changes, the effect of nudging is almost null concerning the extent of the eastward shift in precipitation, therefore confirming the robustness of our previous analysis.

[Figure]

4. Moreover, the effect of sea surface salinity should also be considered in your discussion or limitations section. Would the increase in SST and a decrease in total precipitation lead to increased salinity levels and how could that potentially diminish evaporation and consequently reduce severe precipitation events?

We will discuss this aspect in the discussion section of the revised manuscript. We thank the referee for pointing that out.

5. How relevant are gradients between the SSTs of the Tyrrhenian and Ionian Sea for the emergence of extreme precipitation events in the region?

Generally, changes in the SST distribution significantly impact the location of cyclone minima over the sea, and gradients between the SSTs of the Tyrrhenian and Ionian Sea could further enhance the emergence of extreme precipitation events in the region. Although particularly interesting, we believe that this question exceeds the scope of this study (it is worth noting that we perform spatially *homogeneous* variations of the SST fields across all domains) and could lead the reader to possible misunderstandings. Therefore, we will perform the analysis suggested for our study period and consider whether to include it in the manuscript or supplementary material in the next steps of the review process.

6. How are the main horizontal wind fields for the precipitation events (Fig. 12) organized? Is there any obvious clustering for the big events, e.g. all originate from the south? Is it possible to annotate the dots of the events with an arrow that shows the direction of the storm path?

We thank the referee for this comment. Adding the direction of the storm paths will not be straightforward, but it would add another interesting piece of information to Figure 12. We will strive to find the most effective way to implement it. Concerning general circulation, broadly speaking, we have two main types of precipitation events, with fronts originating from the Tyrrhenian side (west/northwest) or the Ionian (south), which are typically those with the highest intensity. We provided a comprehensive study on weather patterns in the Mediterranean and their impact on the Calabria region in Mastrantonas et al. (2022).

7. Are the obtained spatial shifts of the event centers consistent for small perturbations (e.g. another PBL scheme or varied initial conditions)? Was there any spin-up performed to exclude impacts of imbalanced soil moisture?

We thank the Referee for this comment. About the spin-up: this is a single simulation from September $1^{st}$ to December $31^{st}$, we did not consider the first potential event because it was simulated in the first week of the simulation, as can be observed from Fig. 9, where the event E1 is indeed the second event recorded in the study period. Small perturbations (i.e., using another PBL scheme) will be evaluated only for some events (e.g., event 12) to avoid excessive computational effort, and the results will possibly be shown in the revised version of the manuscript, either in the main text or as supplementary material.

8. How do these extreme events look like in the outer domain: Are the centers of precipitation mass identically located to what was found for the inner domain?

We will systematically analyze all centers of mass for the external domain and provide the results in the next steps of the review process. As a preview, the figure below illustrates the D01 precipitation fields simulated in the three different scenarios (a) SST-1; b) SST0; c) SST+3) for event no. 12. In the outer domain, the main features of the precipitation fields are confirmed, as precipitation shifts eastward and increases with increasing SST.

[Figure]

**Minor**

7. The title is probably not so ideal since your main focus is on SST sensitivity. Maybe better: "… on the Effects of Mediterranean Sea Warming"?

We thank the Referee for pointing that out. We will modify the title according to the Referee's comment, and following the same suggestion provided by Referee #3.

8. L33: Give the actual years instead of "In the last two years"

We will change the manuscript according to the comment and make the years 2023 and 2024 explicit.

9. L324: From Fig. 13 it seems that the storm system of event 12) travels along the coast in northward direction and that the high precipitation over the sea occurs after it traveled over the east part of the Calabrian coast rather than "exploding before".

We thank the Referee for pointing that out. We were referring to exploding before reaching the Apulia (north) coastline, but we agree the sentence is misleading. We will clarify this aspect in the revised manuscript.

10. L391: I think the investigation should not be called "comprehensive" since many real-world aspects had been left out in this PGW experiment.

The Referee is right, because we focus on SST increase. We will smooth such a sentence following the Referee's suggestion.

11. Figure 2: add the term ERA5-Land to the caption.

The term will be added in the caption. Furthermore, the figure will be slightly modified in accordance with Major Comment No. 2 of Referee #2.

12. Figure 3: add the term ERA5-Land to the caption. Increase image resolution.

The term will be added in the caption. Furthermore, the figure will be modified in accordance with the specific comment about Figure 3 of Referee #2.

13. Figure 4: "Mann-Kendall and Sen's slope test for observations of a) ..."

The caption will be modified accordingly.

14. Figure 5: Increase image resolution. It would be good to scale the point size by the trend values. It's also hard to distinguish the significant values. A different color might be better to show them. Add "observations" to figure caption.

According to this comment and another from Referee #2 (specific comment related to Fig. 5), Figure 5 will be modified as shown below. Stations with significant values for both trends will have a thicker black border. Furthermore, the caption will be modified as suggested.

[Figure]

15. Figure 7 & 8: annotate the periods in the figure and add "GCM ID" or similar to the y-axis.

Concerns about Figs. 7 and 8 were also raised by other referees (Referee #1, Results and Discussion comment no.6, Referee #2, comment no.5, and Referee #3, comment no.5). Following Referee #2's suggestion, the representation of projected SST increase will be changed entirely, using only one multipanel figure with four spaghetti graphs, showing the SST increase compared to 1985-2014 in the periods 2040-2069 and 2070-2098, considering SSP3-7.0 and SSP5-8.5 scenarios, respectively. Additionally, according to Referee #1's comment, the area on which we will base our calculations is no longer the entire Mediterranean basin, but the external domain D01. The new figure is shown below (in the spaghetti graphs,

the dotted line represents the median behavior). We observed a slight increase in projected SST. Further details will be provided in the revised text.

[Figure]

16. Figure 10: Add "precipitation" somewhere in the figure caption

We will modify the caption as suggested.

17. Figure 13: Add red cross section line also to c) and e) and add the center of mass points for event 12; what is the unit of Omega?

We will add the red cross-section line and the center of mass points shown in Figure 12 in the sub-figures. Omega is formally defined as the variation of pressure over a time interval and is calculated in $hPa\ h^{-1}$. We will add units to the color bar legend.

**References**

Beranová, R., R. Huth, & V. Vít (2025). A multi-dataset analysis of precipitation trends in Europe. J. Hydrometeor., https://doi.org/10.1175/JHM-D-24-0114.1, in press.

Mastrantonas, N., Furnari, L., Magnusson, L., Senatore, A., Mendicino, G., Pappenberger, F., & Matschullat, J. (2022). Forecasting extreme precipitation in the central Mediterranean: Changes in predictors' strength with prediction lead time. Meteorological Applications, 29(6), e2101.

---

## Author Comment (AC3)

We would like to warmly thank the Referee for their highly detailed review of our paper. The comments and suggestions provided will contribute significantly to improving the quality of our paper, making it more effective and clearer. Please find below our point-by-point responses (in red text).

**General Comments**

The manuscript "Increasing Daily Extreme and Declining Annual Precipitation in Southern Europe: A Modeling Study on the Effects of Mediterranean Warming?" by Senatore et al. addresses an important and timely issue of the apparent paradox between increasing event (or daily) rainfall amounts despite the overall drying of the Mediterranean. While this issue was previously discussed in many studies, at least since Alpert et al. (2002), a complete answer seems to be lacking from the literature, and studies that provide pieces of this puzzle are needed.

The authors use long-term ERA5-Land data to show the context of precipitation pattern change throughout the Mediterranean (and Europe), numerous gauge records from Calabria to show these changes over the region, and high resolution WRF model simulations to investigate the impact of changing sea surface temperature on precipitation. The combination of tools and analyses provides a potentially valuable contribution to the field; however, while the study, as it is contextualised, is ambitious, it suffers from several weaknesses that make this framing exaggerated. Specifically, (a) the manuscript lacks clear articulation of the knowledge gap it seeks to address (is it precipitation increase during extremes while total precipitation decreases? Is this clearly answered in the text?); (b) the authors invest significant effort in justifying the representativeness of both the Calabria region and the specific season chosen for simulation, suggesting that these are indicative of broader Mediterranean conditions and climate change trends. While these arguments are reasonable, the paper would remain valuable even if it were framed more narrowly — as a regional case study focused on Calabria and/or a specific season. Overextending the generalization to the entire Mediterranean may risk overstating the broader applicability of the results; (c) Lastly, the authors frame the research in the intensification of extreme precipitation while overall precipitation is decreased. However, their results suggest increase in total precipitation. While these are reasonable results given the framework of increased SST only, this should be explicitly mentioned, and the authors should explain whether their study shows an exception to the general behaviour, or it only addresses a specific part of climate change effects on precipitation.

Given these comments, in my view, the manuscript should undergo major revisions before it could be published in HESS. Further comments are written below.

We thank the Referee for the generally positive feedback. We particularly appreciate their acknowledgment that, about the issue addressed, "a complete answer seems to be lacking from the literature, and studies that provide pieces of this puzzle are needed", which provides strength to the general purpose of our paper, and the potential added value of the combination of tools and analyses used. Concerning the three general comments that have arisen:

  (a) The main aim of this paper is to investigate the extent to which Mediterranean Sea warming contributes to the seemingly counterintuitive increase in daily precipitation extremes in southern Europe, despite a general decline in annual precipitation. We acknowledge that this main purpose could be more clearly addressed (as it has also been highlighted by other referees, e.g., Referee #1), and to this aim, we will reshape the Introduction section, taking up also the Referee's suggestions and comments

(particularly, major comment no. 1). We firmly believe that the text contains useful insights concerning the topic, and will try making our answer clearer.

(b) We acknowledge the recalled risk of "overextending the generalization to the entire Mediterranean", which may overstate "the broader applicability of the results." On the other hand, we do not consider it correct to overlook the representativeness of the study area in the broader context of the Mediterranean, which, as the Referee states, is supported by reasonable arguments. Therefore, in the revised version of the manuscript, we will strive to balance these two aspects, defining the limits and boundaries of the results obtained more clearly.

(c) We assume that the conclusion of the Referee that our results "suggest increase in total precipitation" lies in the fact that, considering the precipitation events analyzed, the total amount is higher with SST+3 than with SST0. However, in the manuscript, we stated that "each event was considered as standing alone with respect to others" (L179) and "can be considered as isolated" (L347), while, among the information missing in our PGW scenario, "the expected frequency of cyclones, given the large-scale circulation dynamics changes induced by climate warming, is another relevant piece of information missing" (L349). Therefore, while we have proven that most of the events, individually analyzed, produce more rain under SST+3 conditions than under SST0, we can't prove a generalized increase in total precipitation because we can't provide information about the projected frequency of such events (and about other aspects, as pointed out by the Referee in their major comment no. 8). In the revised manuscript, we will include this discussion.

Finally, below we reply to the major and specific comments of the Referee.

**Major comments**

1) The last paragraph of the introduction provides both further motivation and background for the study (L64-67) and the task/object of the manuscript (L67-L84). In my view, the introduction lacks a clear description of the *need* for the work. After reading the first part of the introduction readers are likely familiar with some of the challenges and background related to understanding the impact of global warming on precipitation in the northwest Mediterranean. However, there is no direct statement clarifying what is not known and what is going to be resolved with this work. Please consider adding a description of the *need* and separate it from the *task* and *object*.

We thank the Referee very much for this valuable suggestion. As stated before (reply to general comment a), and as replied to other referees (particularly, Referee #1), in the revised Introduction we will strive first to clarify the research gap we aim to fill (the *need*) and then what results we aim to achieve and how (the *object* and the *task*), highlighting the peculiarities (strengths) of the analysis performed.

2) The use of EURO-CORDEX in the study is clear. However, presenting the results over the entire domain is not necessary. The authors should consider focusing this part of the results on the Mediterranean area only.

Thank you for this suggestion. Although we were somehow "charmed" by the latitudinal gradient achieved over Europe, especially concerning PRCPTOT trends, we agree that an area of interest more focused on the Mediterranean Basin can contribute to a more robust and understandable storyline. Furthermore, limiting the study area also helps in comparisons with

observations. Figure 2 of the manuscript will be replaced with the one below (upper panel for PRCPTOT and lower panel for RX1day). Additionally, Figure 3 (which will be further modified in accordance with other comments - e.g., the Referee's specific comment on Figure 3) will display the same spatial extent.

[Figure]

3) The Calabria region is at the focus of this study. The introduction lacks clear presentation and reader guidance about the region and why it serves as a case study, or why it is interesting to a broader readership unfamiliar with it.

We will emphasize the importance of that study area in the context of the Mediterranean Basin and the selected period simulated in our high-resolution modeling framework, both in the Introduction and Data and Methods sections.

4) The comparison of ERA5-Land and gauge data seems insufficient to conclude that ERA5-Land "can catch the contrasting daily and annual precipitation trends" as well as the "results achieved along the entire northern Mediterranean coast". This is because: (a) no direct comparison was made. Could you provide a direct comparison of the results? For example, a scatter plot of gauge vs. ERA5-Land trends. This could either be done on a gauge-pixel basis, or interpolated gauge-to-pixel-pixel basis. (b) The fact that results are promising over one area (Calabria) does not mean they are like that, or should be like that, over extensive areas such as the northern Mediterranean. If this claim is something you would like to present, please show a wider comparison using gauges from other areas of the northern Mediterranean as well. Daily gauge records are rather easy to find, and thus, this analysis seems manageable. Similar comparisons could be made with other, gridded datasets, considering their limitations (e.g., L228-239).

While the analysis suggested in (a) can be easily performed, that suggested in (b) goes well beyond the scope of our paper and requires a specific study devoted to it. We claimed that the results in Calabria "further strengthen the reliability of the results achieved along the entire northern Mediterranean coast", meaning that Calabria is a case in which the reliability of the ERA5-Land dataset in reproducing trends is demonstrated. We acknowledge that this sentence is misleading and will be modified. Additionally, we will provide more literature results, such as those by Gomis-Cebolla et al. (2023), Oukaddour et al. (2025), and the very recent Beranová et al. (2025), and investigate the possibility of using the recently published dataset from Vicente-Serrano et al. (2025) for comparison, specifically concerning annual precipitation.

As a preview of what will be shown in response to (a), below the overlays of the Sen's slope extracted from both datasets for PRCPTOT (upper panel) and RX1day (lower panel) are shown. Of course, the gauges capture more localized variations (not necessarily representative of an entire ERA5-Land grid cell), and the ERA5-Land dataset provides more smoothed precipitation time series, suggesting the potential for a hybrid approach in precipitation monitoring, which utilizes both gauge data for localized accuracy and gridded data for regional assessments.

[Figure]

[Figure]

5) Sect. 3.2 and Figures 6 and 7: Given that there is no description of the use of specific CMIP models over specific months, in my view, the details provided in Figures 6 and 7 are excessive. You could provide, for example, a plot like the one below, showing the changes throughout the year. If the specific models are needed you could plot them as spaghettis, and if they are not needed, a shaded area would be more helpful. This could be done for the actual values and for the difference ("delta") values, and if it looks alright you could even provide the two SSP scenarios on the same plot, thus, minimising the figures to only two panels. More generally, it seems like the whole purpose of this section is to justify the use of the specific numbers which were then simulated, which is alright, but you could provide these numbers with a symbol on a graph (e.g., add the 2019 values and the -1, 0 and +3 values to the "delta" panel plot) and make the text much shorter (one paragraph).

[Figure]

We sincerely thank the Referee for this suggestion. This concern was also raised by the other referees (Referee #1, Results and Discussion comment no.6, Referee #3, comment no.5, and,

partially, Referee #4, minor comment no. 15). Following Referee's suggestion, the representation of projected SST increase will be changed entirely, using only one multipanel figure with four spaghetti graphs, showing the SST increase compared to 1985-2014 in the periods 2040-2069 and 2070-2098, considering SSP3-7.0 and SSP5-8.5 scenarios, respectively. Additionally, according to Referee #1's comment, the area on which we will base our calculations is no longer the entire Mediterranean basin, but the external domain D01. The new figure is shown below (in the spaghetti graphs, the dotted line represents the median behavior). We observed a slight increase in projected SST. Further details will be provided in the revised text.

[Figure]

6) WRF event simulation validation. In the FSS analysis, you use the 95th percentile, which is quite high, and get rather low values. While you mention that 0.5 "typically indicates valuable skill", actually it is 0.5 + the occurrence of half of a random forecast (Roberts and Lean, 2008), meaning that almost none of your forecasts are "useful" as termed by Roberts and Lean (2008). Even if we take the small portion of useful forecasts, they lie mainly in the larger averaging domains (e.g., "Size 19"), so the forecast is useful for 180X180 km and more, which is quite a bad statement. Please consider using the FSS with a lower threshold, unless you're specifically interested in pinpointing the really high precipitation accumulation values. You could, for example, take the 50th percentile out of the "rainy" pixels (e.g., where rainfall accumulation is greater than 1) or something similar. That's just an example for a threshold that I see as valuable but other thresholds could also be used. The thing is that taking the 95th percentile means you are trying to forecast extreme values and you actually show that this prediction is bad. Additionally, when describing FSS and CSI scores for the SST-1 and SST+3 simulations you are showing validation of events which we do not expect to be similar to the observed. Therefore, the meaning of the validation of this forecast is vague. You can, however, suggest that lower scores indicate rainfall patterns had changed between the events, but this is sort of a weird way to examine it. A simpler way would be an application of a metric that measures similarity between precipitation fields, for example the SAL analysis (Wernli et al., 2008) where SST0 serves as the "truth" while each of the other scenarios is examined against it.

We thank the Referee for pointing out several aspects of the verification strategy. Our statement of FSS value greater than 0.5 could be misleading, as it should be a valuable skill above 0.5 + the occurrence of half of a random forecast. Indeed, how to use the FSS at its best is an issue still under investigation (e.g., Antonio and Aitchison, 2025). Following the Referee's suggestion, we have relaxed the percentile thresholds by using the 90[th] percentile, which was also employed by Senatore et al. (2020) and Antonio and Aitchison (2025). However, since we consider 20 different events, we will just highlight the 0.5 threshold by inserting a dashed line in the new figure (which is shown below). The revised figure highlights that even with relatively small window sizes (i.e., not only size 19), SST0 forecasting is skillful, and the importance of both proper SST and high-resolution representations is also critical (i.e., results with SST-1, SST+3, and ERA5-Land are poorer).

[Figure]

While we believe it makes sense showing SST-1 and SST+3 results compared to observations (for the reasons exposed above), taking up the Referee's suggestion, we conducted also a SAL analysis (Figure below, where the dashed line indicates the median, while the dotted lines highlight the 1[st] and 3[rd] quartiles, respectively). Since the reference simulation is the SST0, and this analysis highlights the effect of increasing or decreasing the SST, we will add such a graph to Figure 12 to emphasize the effect of the SST scenarios.

[Figure]

7) L327–337 and Fig. 13: Event #12 shows quite a lot of precipitation over the Tyrrhenian Sea and even more over the Calabrian peninsula. From the accumulated precipitation map (Fig. S12) it seems like there is a significant orographic enhancement of rainfall during the event. At the same time, it seems like there is quite a low correlation between the CAPE values presented (from the night before the peak of the event) to the accumulated precipitation map. This requires elaboration: either the night before is not a good representative time step, or CAPE is not such an important indicator for precipitation during the "actual" event. If the latter is the case, why should CAPE be a good indicator for the changes occurring between the three different scenarios? Did you consider examining the moisture flux perpendicular to the coastline/mountain ranges? Could you think of other mechanisms explaining this shift towards the Ionian Sea? How about the rainfall across the entire domain, does it go up or down in this event? A graph would help, so please consider adding another panel to Fig. 11 with precipitation over the entire domain. Can this shift be arbitrary? From Fig. 12 it seems like the shift is consistent, but the explanation is not clear to me. Can it be related to a shift of the cyclone center toward the Ionian Sea? If so, why? In my view, to better explain this part, a more detailed analysis is needed. Last note about this topic, "Negative values of omega, related to vorticity advection (e.g., Lenderink et al (2017))" — such values are not necessarily the result of vorticity advection, and are important indicator for precipitation even without vorticity advection. Also, if I remember correctly, the vorticity advection topic is not discussed in Lenderink et al (2017).

We agree that the orographic enhancement is a key factor in the precipitation development for event #12. We believe that the increasing CAPE shown in Figure 13 effectively highlights both the thermodynamical and dynamical aspects of our simulation results, emphasizing the increasing atmospheric instability over the Ionian Sea, with the humid air mass coming from the south and being fed by the increasingly warmer sea surface, changing significantly the cyclone track and the spatial pattern of precipitation. We first reported this result in a paper published more than 10 years ago (Senatore et al., 2014). However, we will strive to strengthen our findings further and explore the suggested analyses by computing the vertical moisture flux perpendicular to the mountain ranges for the three scenarios. Concerning rainfall across the entire domain, of course, it also increases for event #12, as is the case for all the analyzed events. This outcome is already shown in Table A1 and further illustrated in the figure below (which will be included in the paper, as suggested).

[Figure]

Finally, regarding omega, we agree with the reviewer that the phrase "related to vorticity advection" is misleading and will be removed. Additionally, we agree that the reference to Lenderink et al. (2017) is incorrect in this context.

8) Increase in extreme precipitation despite overall drying. The authors claim to investigate the intensification of heavy precipitation in the context of overall drying over the Mediterranean. However, the analysis seems to show that precipitation is increased across almost all rain events within the examined season in SST+3 and the opposite for SST-1. This raises questions about the apparent contradiction between their results and the broader drying trend they show and refer to. I would like to see some discussion explaining why this is the case and whether the results actually show the intensification of extremes, or rather they show a general intensification which is related to rising SST. If the latter is the case, this should be explained, and the discussion could describe the contradiction by suggesting alternative reasons for the overall drying, like reduced cyclone frequency (Zappa et al., 2015), shortening of the rain season (Hochman et al., 2018), decreasing land-sea gradients (Tuel and Eltahir, 2020) and a decrease in the area and duration of rain events (Armon et al., 2022).

This comment appears to be related to the general comment (c). We reiterate here that, in the manuscript, we stated that "each event was considered as standing alone with respect to others" (L179) and "can be considered as isolated" (L347), while, among the information missing in our PGW scenario, "the expected frequency of cyclones, given the large-scale circulation dynamics changes induced by climate warming, is another relevant piece of information missing" (L349). Therefore, our results support a scenario with "the intensification of extremes", but cannot support "a general intensification which is related to rising SST". Besides the reduced cyclone frequency, all the alternative/complementary reasons listed by the Referee, accompanied by the respective literature references, are valuable suggestions for us to enhance our discussion.

**Specific comments**
L18-L28: The first paragraph is very general on the one hand, but puts a lot of emphasis on the expansion of the Hadley Cell on the other hand. While this is nice as a general introduction, it does not fully lead the reader to a better understanding of the background related to this specific study. I would suggest either being more specific — i.e. what is special about the climate change of the Mediterranean, or adding other examples to what governs climate change effects over the region in addition to the expansion of the Hadley Cell.

We thank the reviewer for this comment. While we (as suggested) will keep the issue of the expansion of the Hadley Cell, we will be more specific about climate change issues in the Mediterranean Basin.

L34-36: The claim for increased cyclonic activity is **not supported** by the cited reference. Aragao and Porcu's (2022) claim is that their algorithm produces 40% more cyclones compared with other cyclone detection algorithms. Please revise. Please correct also the connection to medicanes. Medicanes are indeed formed in the Mediterranean and lead to destruction, but this is not related to the fact the number of cyclones in the Mediterranean is increasing (which, in any case, is not supported by the reference cited).

We thank the Referee for this very timely comment. The reference to Aragao and Porcù (2022) will be removed, and the entire sentence rewritten.

L105-106: Did the authors apply the mentioned techniques? If so, would you please elaborate? If not, please describe better the procedure and mention who is responsible for it. Could this procedure change the value of extreme events?

Yes, we applied the mentioned techniques to estimate missing values and reconstruct detected outliers in a large dataset concerning both temperature and precipitation in Calabria. To be precise, concerning precipitation, we only used linear regression; therefore, the text will be modified accordingly. The technique was applied as follows:
- First, daily rainfall data were acquired for each station from 1955 to 2023. If less than 15 days per year were missing (approximately 4%, such as proposed in several studies, e.g., Aguilar et al. 2005, Donat et al. 2013 and Stephenson et al. 2014) the PRCPTOT and RX1day values were calculated for that year.
- If more than 15 days of data were missing for a given station in a specific year, we performed a linear regression between the available data of that station and those of a nearby station whose data were highly correlated (in all cases, we achieved $r$ values greater than 0.8, often much greater). Then, we used the linear regression equation achieved to fill the gaps.

Although this procedure could potentially influence the values of extreme events, it was approached with careful consideration of the data's characteristics.

L123: To my view, there is not much of a comprehensive overview of the dataset in Table 1. Rather, there is a list. Please delete the text in the parentheses except for the words "Table 1".

We will modify the manuscript according to the Referee's comment.

L133-134: " Non-parametric trend tests like the... were employed" — please be specific. Are these the only tests or are there others?

We used only these tests. The sentence will be modified to make the text clearer: "The non-parametric trend tests of Mann-Kendall (identifying significance at a 5% level) and Sen's slope estimator (determining the trend slope per year) were employed to analyze…"

L195 (and L208): " resulting in more probable flooding challenges across Europe (Fig. 2b)." This claim is problematic since greater RX1day does not necessarily mean more probable flooding challenges, as floods come in different flavours. See for example Bloschl et al. (2019).

We understand the point raised by the Referee and will make these sentences smoother. Regarding Bloschl et al. (2019), it is noteworthy that the analysis is limited in several Mediterranean areas due to the (sadly chronic) lack of observations. E.g., the graph of the uncertainty of the river flood discharge trends in terms of standard deviation (Extended Data Fig. 2b) exhibits a sharp latitudinal gradient in Greece and southern Italy.

L196: " These results are largely consistent with previous literature." Is there literature showing ERA5-land trends over the EURO-CORDEX region? If there is, this should have been mentioned in the introduction. If there is not, it is better to explicitly mention the "previous literature". In that case, the introduction misses a description of what was not done before which you are showing here (e.g., calculating trend in ERA5-Land versus trends in other models).

We referred to previous literature regarding the trends achieved, either by using or not using ERA5-Land. We considered the reference to the 6[th] IPCC Assessment Report sufficient since it is a sort of summary of many other works. However, we will contextualize the sentence better, both here and in the introduction. Interestingly, a very recent paper from Beranová et al. (2025) provides results very close to ours concerning winter and summer precipitation trends in Europe in the period 1961-2010 (by the way, another novelty of our research is the significantly updated dataset used, with 2023 as the final year).

Figure 3: Could you add axes labels to the small quadrant graph? This would make it much easier to read (rather than remember) what every zone represents. A different approach would be to incorporate all zones into one graph but vary the colours. Could you try this approach in order to minimise the number of panels for this figure?

We sincerely thank the Referee for this brilliant idea. Figure 3 will be updated as shown below, with different colours representing the four quadrants. Additionally, as noted in comment no. 2, we will focus on the Mediterranean. Finally, a specific figure in the Methods section will be added to explain the meaning of the four zones (please refer to our reply to Referee #1, Data and Methods comment no. 2).

[Figure]

Figure 4: The resolution of the figure is too low. Please provide a better resolution version of it.

Figure 4 will be modified and enhanced with improved resolution, as depicted below (left panel for PRCPTOT and right panel for RX1day trends in terms of Sen's slope).

[Figure]

L213-218: The results of significant trends in RX1day should be treated carefully! Considering random processes, we would expect to find more or less 7 gauges (5%) with positive and 7 gauges with negative trends. Please make sure to address this. In contrast, L219-227 discuss mainly the non-significant trends. There, a note should be highlighted saying that these are non-statistically-significant results.

We agree with this comment. The text in LL213-227 will be revised to address the issue of the low number of RX1day significant trends and the fact that mostly non-significant trends are discussed.

Figure 5: I would suggest the same as in Fig. 3 — combine all quadrants and vary the colours.

Similar to the comment related to Figure 3, Figure 5 will be modified as follows.

[Figure]

L240-241: This claim is not clear to me. Please explain.

This comment appears to be related to the general comment (b). The statement emphasizes the representativeness of the study area within the broader Mediterranean context, which, as the Referee states, is supported by reasonable arguments. Furthermore, we highlight the availability of a dense and reliable monitoring network. In other words, this region can provide insights into how local climatic conditions can impact broader regional trends.
This statement will be revised and further clarified in the manuscript, in accordance with our reply to the general comment (b).

L254: Please explain what is the value which the "±" sign refers to (is this the range, standard deviation, 10–90 quantile range?)

The values reported with the symbol ± represent the standard deviations for both the SSP3-7.0 and the SSP5-8.5 scenarios. We will clarify this in the revised manuscript.

Figure 9 and L276–277: Could you provide a reference to another figure, preferably Fig. 1, where you show the area over which you do this spatial interpolation and its boundaries? Is it the same area for the observations and model? What type of interpolation are you using?

We interpolated over the Calabria region boundaries, shown in Figure 1a, using the 150 rain gauge stations highlighted. The description of the methodology used for spatial interpolation is given in LL110-113 and involves IDW (Inverse Distance Weight). We will include a reference in the revised manuscript to the section mentioned above.

Additionally, some events seem to have a less skilful simulation, especially in terms of the spatial distribution of rainfall, for example, event #2. Could you provide some details why that is the case while others are simulated almost perfectly?

If we have understood the question correctly, a detailed answer would likely exceed the scope of this paper. In this context, we can provide some general reasons, including the correctness of the boundary conditions, the difficult predictability of local convective events, and the overestimation of the orographic effect by WRF simulations in certain situations (in the case of event #2 it seems to occur in the central-southern part of the region). Furthermore, in the case of event #2, the Calabria region was at the very border of the transit of a cold front. In any case, we prefer not to distract the reader from the paper's main topic.

Figure 10. If I understood you correctly, boxplots represent the variability across the 20 different events. Is that correct? Please describe this in the text. Furthermore, the labels "Size1... Size19" are not clear. Is this when averaging across 1 pixel, 3 pixels... 19 pixels? Please explain.

The Referee understood correctly that the boxplots represent the variability of the 20 different events. As far as the FSS is concerned, the size represents the number of cells considered in the moving windows. In the revised manuscript, this information will be included in the figure caption. Regarding the SAL, we used a violin plot, which will be added to Figure 12, as it highlights the difference between the SST scenarios and SST0, as suggested in major comment no. 6 by the Referee.

Figure 11. What is symbolised by the shaded area? What is represented by the line? Please explain. Also, could you add a panel showing the (non-normalised) linear regressions discussed in L299–L310? Additionally, when presenting values in a normalised axis, it is common to present it in log distances, such that e.g., a decrease of 50% has the same distance from 0 change as an increase of 100%.

The shaded area represents the 95th percentile confidence level performed for the 2nd-order polynomial fitting (i.e., the line) over the two different scenarios. In LL299-310, only one couple of linear regressions is discussed (L300). Probably the Referee also refers to the other two couples of regressions (L297 and L318). We feel that this panel could be redundant, given Table A1. However, we will consider the opportunity to add the graphs. Regarding the representation in log distances, we've made various attempts to improve the visual format of the graph and concluded that log distances do not add any benefit (i.e., they do not highlight any specific feature) while making the graph less readable.

L342–343: Could you explicitly write what you mean by those features? The authors' analysis is focused on precipitation accumulation, rather than intensities, and " tracking" is not clear to me in this context. "Cyclone tracks" is further mentioned in L401, which is again, not clear to me.

We agree that the terms used may be misleading and imprecise in this context. We referred to "peak intensity" because the intensity of the precipitation is reasonably very high in the center of mass of the accumulated precipitation (but not necessarily the highest), and to "tracking" because the locations of the centers of mass of the accumulated precipitation are related to the modified cyclone tracks. We will rewrite these sentences in a more precise way.

L355–357: Similar conclusions were made earlier by Zappa et al. (2015). Please consider citing them.

We thank the Referee for this valuable suggestion. We will add the reference in the revised manuscript.

L387: For Storm Daniel, please consider using more relevant literature focusing on the meteorology and hydrology of the event, e.g., Armon et al., (2025) or Flaounas et al. (2024).

Once again, we thank the Referee for their valuable suggestions. We will revise the suggested references and add them to the revised manuscript.

L405–407: This was shown before, over many different studies (e.g., Prein et al., 2015; Ban et al., 2014; Pichelli et al., 2021; Coppola et al., 2020).

We agree that the sentence, in the way it is written, could sound a bit trivial. On the other hand, the topic remains currently highly debated (e.g., Soares et al., 2024; Fosser et al., 2024). Furthermore, we noticed that the main features of precipitation change can be observed already by examining the outer (lower resolution) domain (reply to Referee #4, comment no. 8). We will revise the sentence to take into account new insights, contextualize it better, and consider the suggested references.

L411: Since many studies have already used PGW with more parameters changed rather than only SST, this sentence is not clear to me. Please consider revising this statement. Also, the statement in L413–415 is not in place here; it should either go in the discussion or be deleted, because other factors are competing with it, such as soil drying because of longer dry spells.

We will revise this sentence accordingly. Concerning this point, useful insights are also provided by Referee #4 (comments nos. 2, 3, and 4). However, it is also of interest to us to refer to "complete" global warming scenarios to provide the whole picture, as we are currently working on convection-permitting climate simulations in the study area (https://doi.org/10.5194/egusphere-egu25-15936), which represent an area of future investigation for us. The sentence at LL413-415 will be removed.

Data availability: The netcdf files contain accumulated precipitation, but the 'Times' (at least for the SST+3) vector is corrupted (it contains '0' only). Additionally, there is no spatial information except for the number of grid cells. If you can, it would be very helpful to accommodate the real times in the vector and add information about the coordinates of the data, e.g., lat/lon vectors or arrays.

We thank the Referee for noting that and will add the time strings and the lat/lon arrays in the data uploaded to the Zenodo repository.

**Technical corrections**

L40: "contributes to lead" — please stick with either contributes or lead.

We will revise the manuscript according to the Referee's comment, and use lead.

L42: "trigger dynamically" — please revise. You could, e.g. use "dynamically interact with orographic lifting" or something similar.

We will modify the manuscript accordingly.

L60: "as the Ianos cyclone occurred in 2020, producing" would be more readable if you expand the sentence like "such as cyclone Ianos that occurred in 2020, which produced..."

We will modify the manuscript accordingly.

L76: "events occurred" please expand to "events that occurred".

We will modify the manuscript accordingly.

L97: "lies" should be "that lies".

We will modify the manuscript accordingly.

L295: "evapotranspiration" should probably be "evaporation".

We thank the Referee for pointing out the typo. We will modify the manuscript accordingly.

Fig. 12: Please correct the "Tyrrenian" label in Fig. 12 to be similar to what's written in the text i.e., "Tyrrhenian".

We thank the Referee for noting that. We will modify the figure accordingly.

**References**

Aguilar, E., Peterson, T. C., Obando, P. R., Frutos, R., Retana, J. A., Solera, M., ... & Mayorga, R. (2005). Changes in precipitation and temperature extremes in Central America and northern South America, 1961–2003. Journal of Geophysical Research: Atmospheres, 110(D23).

Antonio, B., and L. Aitchison (2025). How to Derive Skill from the Fractions Skill Score. Mon. Wea. Rev., 153, 1021–1033.

Beranová, R., R. Huth, & V. Vít (2025). A multi-dataset analysis of precipitation trends in Europe. J. Hydrometeor., https://doi.org/10.1175/JHM-D-24-0114.1, in press.

Donat, M. G., Alexander, L. V., Yang, H., Durre, I., Vose, R., Dunn, R. J., ... & Kitching, S. (2013). Updated analyses of temperature and precipitation extreme indices since the beginning of the twentieth century: The HadEX2 dataset. Journal of Geophysical Research: Atmospheres, 118(5), 2098-2118.

Fosser, G., Gaetani, M., Kendon, E.J. *et al.* (2024). Convection-permitting climate models offer more certain extreme rainfall projections. npj Clim Atmos Sci 7, 51.

Gomis-Cebolla, J, Rattayova, V., Salazar-Galán, S., Francés, F. (2023). Evaluation of ERA5 and ERA5-Land reanalysis precipitation datasets over Spain (1951–2020), Atmospheric Research, 284, 106606.

Oukaddour, K., Fakir, Y., & Le Page, M. (2025). Assessment of five global gridded precipitation estimates over a southern Mediterranean basin (Tensift, Morocco). Geomatics, Natural Hazards and Risk, 16(1), 2468850.

Senatore, A., G. Mendicino, H. R. Knoche, and H. Kunstmann (2014). Sensitivity of Modeled Precipitation to Sea Surface Temperature in Regions with Complex Topography and Coastlines: A Case Study for the Mediterranean. J. Hydrometeor., 15, 2370–2396.

Senatore, A., Davolio, S., Furnari, L., and Mendicino, G. (2020). Reconstructing Flood Events in Mediterranean Coastal Areas Using Different Reanalyses and High-Resolution Meteorological Models, Journal of Hydrometeorology, 21, 1865–1887.

Soares, P. M. M. et al. (2024). The added value of km-scale simulations to describe temperature over complex orography: the CORDEX FPS-Convection multi-model ensemble runs over the Alps. *Clim. Dyn.* 62, 4491–4514.

Stephenson, T. S., Vincent, L. A., Allen, T., Van Meerbeeck, C. J., McLean, N., Peterson, T. C., ... & Trotman, A. R. (2014). Changes in extreme temperature and precipitation in the Caribbean region, 1961–2010. International Journal of Climatology, 34(9), 2957-2971.

Vicente-Serrano, S. M., Tramblay, Y., Reig, F., González-Hidalgo, J. C., Beguería, S., Brunetti, M., ... & Potopová, V. (2025). High temporal variability not trend dominates Mediterranean precipitation. Nature, 639(8055), 658-666.

---

## Author Comment (AC4)

We would like to warmly thank the Referee for their thorough review of our paper. The comments and suggestions provided will contribute significantly to improving the quality of our paper, making it more effective and clearer. Please find below our point-by-point responses (in red text).

This is an interesting study on understanding better the precipitation paradox in the Mediterranean (decreased mean precipitation/increased extremes). It falls within the scope of HESS, and I believe it could attract the interest of the scientific community. While there are different approaches and sound datasets used, there are no straightforward linkages between the various parts of the analysis.

We thank the Referee for the positive feedback and acknowledge that further effort is needed to harmonize different parts of the analysis (and of the manuscript). Below, we attempt to address the suggested concerns and comments.

1. The overall presentation is well structured and clear. One exception is the inclusion of discussions in the results. I would include any discussion in a separate section or present it with the conclusions. High-quality visualizations are used for the presentation of results. Some more effort could be put into making the language style more fluent.

We will better separate results and the discussion, devoting a specific section to the latter. Indeed, Section 3.4 is already a discussion section, specifically addressing certain issues. We will restructure and expand it.

2. In the title, please mention that you refer to the "Mediterranean Sea Warming".

We agree with the Referee's comment and will modify the title accordingly; moreover, the same suggestion was provided by Referee #4 in comment no. 7.

3. Overall, the introduction section is informative, however, the list of references is not exhaustive. Topics such as future extreme precipitation trends in the Mediterranean or why the region is characterised as a climate change hotspot could be better covered. Any research gaps and main objectives of the present analysis could be more emphasized.

We agree that the introduction section could be improved with references specifically addressing the topic discussed in the paper. Indeed, we considered several of these references in the Section 3.4 (Comparison with previous studies), but we will consider to cite at least part of them in the introduction, as well as to add other references, e.g. the quite famous paper of Giorgi (2006), stating the Mediterranean as a prominent hotspot due to amplified warming and precipitation decline, and other references on regional future extreme precipitation trends (e.g., Zittis et al. 2021; Babaousmail et al. 2020).

4. EURO-CORDEX is mentioned several times in the text, however, the only linkage with this regional initiative is the selection of the domain. Please mention this only once in the definition of your domain of analysis.

The text will be modified according to the Referee's comment, which combines with Referee #2's major comment no. 2, in which it is asked to focus this part of the results on the Mediterranean area only.

5. Since the findings presented in Figures 7 and 8 are mostly used for estimating a reasonable warmer-SST scenario, I strongly recommend moving these two figures to the Appendix. The number of visualizations is already large.

We agree with the Referee. Concerns about Figs. 7 and 8 were also raised by other referees (Referee #1, Results and Discussion comment no.6, Referee #2, comment no.5, and, partially, Referee #4, minor comment no. 15). Following Referee #2's suggestion, the representation of projected SST increase will be changed entirely, using only one multipanel figure with four spaghetti graphs, showing the SST increase compared to 1985-2014 in the periods 2040-2069 and 2070-2098, considering SSP3-7.0 and SSP5-8.5 scenarios, respectively. Additionally, according to Referee #1's comment, the area on which we will base our calculations is no longer the entire Mediterranean basin, but the external domain D01. The new figure is shown below (in the spaghetti graphs, the dotted line represents the median behavior). We observed a slight increase in projected SST. Further details will be provided in the revised text.

[Figure]

6. In the methods section, it is not clear how future SSTs were taken into account in the WRF simulations. Some information is presented in the results (L262-269), however, this approach should be demonstrated in more detail.

In the Data and Methods section, we presented only the datasets used for the SST analysis (LL114-131), including the extent of the area analyzed (L120). Then, in the Results section, we provided more details about the perturbation approach (LL262-269), including the magnitude (from -1 °C to +3 °C compared to current conditions) and the spatial pattern and implementation (e.g., we claimed a homogeneous change). We agree that the SST

perturbation approach can be presented in a clearer and more organic way in the Data and Section method, as pointed out also by the other Referees (e.g., Referee #1 comment "Data and Methods" no. 1), and will change the manuscript accordingly.

7. Some additional explanation of the methods used to derive Figure 12 should also be included in the methods.

We provided details on how we calculated the centers of mass shown in Figure 12 in Section 2.4. We will clarify and expand this section, providing more details about the figure built in the revised manuscript.

8. For increased confidence, I strongly recommend repeating the analysis of Figure 13 for an additional event. For example, for event 15, which is characterised by extreme rainfall, underestimated by the SST0 simulation.

We will follow the suggestion of the Referee and repeat the analysis shown in Figure 13 (which will be expanded according to other referees' comments, specifically Referee #2's major comment no. 7) for event 15 as well. In the next steps of the review process, we will consider whether to include the results in the manuscript or the supplementary material.

**Minor comments** are provided in the attached PDF document.

comment #1 title modification

We will modify the title according to the Referee's comment, and following the same suggestion provided by Referee #4.

comment #2 degrees C per decade is a more common way of presenting temperature trends

We will modify the manuscript according to the Referee's comment and will use °C per decade.

comment #3 Please spell out numbers under 10

We will carefully check all the numbers under 10 and spell them out.

comment #4 This should be defined here.

We will explicitly define the GCM acronym here. We thank the Referee for pointing out that.

comment #5: "on"

We will modify the manuscript according to.

comment #6: anomaly

We will modify the manuscript according to.

comment #7: use plural precipitations

We will modify the manuscript according to.

comment #8: remove some

We will modify the manuscript according to.

comment #9: Please be consistent in the use of units (degrees C or K)

We will use °C and modify the manuscript to ensure consistency in units of measurement.

comment #10: "such as in the case of"

We will modify the manuscript according to.

comment #11: remove "pan"

We will modify the manuscript according to.

comment #12: remove EURO-CORDEX

The text will also be modified according to our reply to comment no. 4.

comment #13: modify "future scenario" to "an increased SST scenario"

We will modify the manuscript according to.

comment #14: remove the first sentence of section 2.1

We will remove the first sentence according to the Referee's suggestion.

comment #15: modify trend analysis to historical trend analysis

We will modify the manuscript according to.

comment #16: remove "of the European Commission"

We will remove this part of the sentence.

comment #17: remove "Skamarock et al., 2021", the WRF extended name and "limited area model"

We thank the Referee for pointing out that. We will remove the citation and modify the sentence according to the Referee's comment since it has already been presented before.

comment #18: Interesting way of presenting results, but should have been explained in methods

The quadrant classification is a method for highlighting the combined trends of two different variables. We already explained it in Methods (LL141-145), but we will strive to explain it more clearly, even with a simple diagram like the one below. However, the representation of the results (Figs. 3 and 5) will be modified according to the comments of other referees (Referees #2, specific comments).

[Figure]

comment #19: I assume that this is the case, but please clarify if these numbers and percentages refer to land-only grid cells.

The achieved results were obtained using ERA5-Land gridded data, which is limited to land areas. This statement will be clarified and pointed out in the manuscript.

comment #20: I would include discussions in a separate section or merge with conclusions.

We thank the Referee for bringing this to our attention. As we stated in our reply to comment no. 1, we will devote a separate section to the discussion.

comment #21: This is not results material.

We agree with the Referee's comment and will move these sentences to the 2.1 section "Dataset and study area".

comment #22: modify "2" to "two"

We will carefully check all the numbers under 10 and spell them out.

comment #23 (Figure 6 caption): modify "yearly" to "annual"

We will modify 'yearly' to 'annual' in the revised manuscript.

comment #24: This is more of an Introduction material

According to the Referee's comment, we will move this part to the introduction section.

**References**

Babaousmail, H., Hou, R., Ayugi, B., Sian, K. T. C. L. K., Ojara, M., Mumo, R., ... & Ongoma, V. (2022). Future changes in mean and extreme precipitation over the Mediterranean and Sahara regions using bias-corrected CMIP6 models. International Journal of Climatology, 42(14), 7280-7297.

Giorgi, F. (2006). Climate change hot-spots. *Geophysical research letters*, *33*(8).

Zittis, G., Bruggeman, A., & Lelieveld, J. (2021). Revisiting future extreme precipitation trends in the Mediterranean. Weather and Climate Extremes, 34, 100380.